# Beef cattle methane emissions measured with tracer-ratio and inverse dispersion modelling techniques

Mei Bai[1], José I. Velazco[2], Trevor W. Coates[1], Frances A. Phillips[3], Thomas K. Flesch[4], Julian Hill[5], David G. Mayer[6], Nigel W. Tomkins[7], Roger S. Hegarty[2], and Deli Chen[1]

[1]Faculty of Veterinary and Agricultural Sciences, The University of Melbourne, Parkville, VIC 3010, Australia
[2]University of New England, Armidale, NSW 2351, Australia
[3]Centre for Atmospheric Chemistry, University of Wollongong, NSW 2522, Australia
[4]Department of Earth and Atmospheric Sciences, University of Alberta, Edmonton, T6G 2E3, AB, Canada
[5]Ternes Agricultural Consulting Pty Ltd, Upwey, VIC 3158, Australia
[6]Agri-Science Queensland, Dutton Park, Qld 4102, Australia
[7]CSIRO Agriculture, Australian Tropical Science and Innovation Precinct, James Cook University, Townsville, Qld 4811, Australia

*Correspondence to* Mei Bai (mei.bai@unimelb.edu.au)

**Abstract.** The development and validation of management practices to mitigate greenhouse gas (GHG) emissions from livestock requires accurate emission measurements. This study assessed the accuracy of a practical inverse dispersion modelling (IDM) ~~micrometeorological~~ technique to quantify methane ($CH_4$) emitted from a small cattle herd (16 animals) confined to a 63 x 60 m experimental pen. The IDM technique calculates emissions from the increase in $CH_4$ concentration measured downwind of the animals. The measurements were conducted for 7 days. Two types of open-path (OP) gas sensors were used to measure concentration in the IDM calculation: a Fourier transform infrared spectrometer (IDM-FTIR) or a $CH_4$ ~~Laser~~laser (IDM-Laser). The actual cattle emission rate was measured with a tracer-ratio technique using nitrous oxide as the tracer gas. We found very good agreement between the two IDM emission estimates ($308.1 \pm 2.1$ (mean $\pm$ s.e) and $304.4 \pm 8.0$ g $CH_4$ head$^{-1}$ d$^{-1}$ for the IDM-FTIR and IDM-Laser, respectively) and the tracer-ratio measurements ($301.9 \pm 1.5$ g $CH_4$ head$^{-1}$ d$^{-1}$). This study ~~shows~~suggests that a practical IDM measurement approach can provide an accurate method of estimating cattle emissions.

**Keywords**: micrometeorological techniques, GHG emissions, beef cattle, spectroscopy, open-path gas sensors

## 1 Introduction

Agriculture is the main source of anthropogenic methane ($CH_4$) emitted to the atmosphere, which includes emissions from ruminants, rice agriculture, waste treatment, and biomass burning (Solomon et al., 2007). Methane is an important greenhouse gas (GHG) with a global warming potential that is 28 times that of carbon dioxide ($CO_2$) in a 100 year time frame (Myhre et al., 2013). Enteric $CH_4$ from livestock is a major source of GHG emissions. A significant effort is being made to mitigate these emissions through diet modification, feed supplements, farm management, grazing strategies, and animal breeding (Min et al., 2020; Vyas et al., 2018); with ruminant nutritional management strategies seen as the most direct impact mitigation option ~~(Cottle et al., 2011)~~(Cottle et al., 2011). Increasingly there is a requirement for mitigation claims to be validated when these practices are applied on-farm (DoE, 2014), and simple and accurate methods for on-farm emission measurements are ~~required~~needed.

On-farm ~~enteric~~CH$_4$ emissions from beef cattle have been measured using three main techniques. 1) Portable respiration hoods for tethered and non-tethered animals (Garnsworthy et al., 2012; Zimmerman and Zimmerman, 2012) directly measure the gas concentration of incoming and exhaust air from individual animals. However, this technique limits the animal's movements, requires intensive training for animals and labor, and it does not account for emissions from the ~~animal~~animal's rectum. 2) Tracer-ratio gas releases from the animal (Johnson et al., 1994), such as SF$_6$ (Grainger et al., 2007), assumes the tracer gas and the emitted CH$_4$ have similar transport paths, so that a tracer measurement can establish the CH$_4$ emission rate. This is a simple technique, but there are challenges with logistics and handling animals that are similar to the respiration hood technique. 3) Micrometeorological techniques are typically considered a herd-scale measurement, where the emission rate is calculated from the measurement of enhanced gas concentrations downwind of an animal herd (Harper et al., 2011), and these include the mass balance technique ~~(Laubach et al., 2008; Lockyer and Jarvis, 1995)~~(Laubach et al., 2008; Lockyer and Jarvis, 1995), eddy covariance (Dengel et al., 2011; Felber et al., 2015), and inverse dispersion techniques (Flesch et al., 2005; Todd et al., 2014). The main advantage of micrometeorological techniques is that they do not interfere with the animals or the environment.

The objective of this study was to examine the accuracy of a practical inverse dispersion modelling (IDM) technique for measuring CH$_4$ emissions from beef cattle. The IDM technique offers the possibility of relatively simple emission measurements, without the need for animal handling or modifying animal behavior. In this study two IDM techniques are used to measure emissions from a small herd of confined cattle, and the results tested against a robust tracer-ratio based measurement.

## 2 Materials and Methods

### 2.1 Experimental design

The study took place at the Chiswick pastoral research laboratory (30° 37' S, 151° 33' E) in Armidale, New South Wales, Australia in February 2013. Methane emissions were measured from 16 Angus steers placed in a temporary 63 × 60 m pen (Fig. 1) located in a flat and open field. There were no other cattle or animal manure storages nearby during the study, and the nearest trees (30 m height) were at least 300 m from the site. Vegetation in the field was removed prior to the study and no pasture was available to graze.

The study cattle had an average body weight of 373 kg (standard deviation = 59 kg). The animals were fed a blended oaten/lucerne chaff ration (90.2% of dry matter, 15.1% crude protein) dispensed from automated feeders (Bindon, 2001) that recorded the individual animal intakes. The ~~feeders~~feeding troughs were cleaned daily, and any remaining feed was weighed to check that the total consumed amount matched the sum of the individual animal intake. Feed and water were offered *ad libitum*. This feeding regime began four weeks prior to the emission measurements. During the seven-day emission measurement period, the average dry matter intake (DMI) was 11.9 kg head$^{-1}$ d$^{-1}$. Cattle manure was not removed during the measurement period. Approximately two weeks before the measurements, each animal was fitted with a backpack (glued to their back) to hold a small nitrous oxide (N$_2$O) gas canister used for the tracer-ratio emission measurements (Jones et al., 2011).

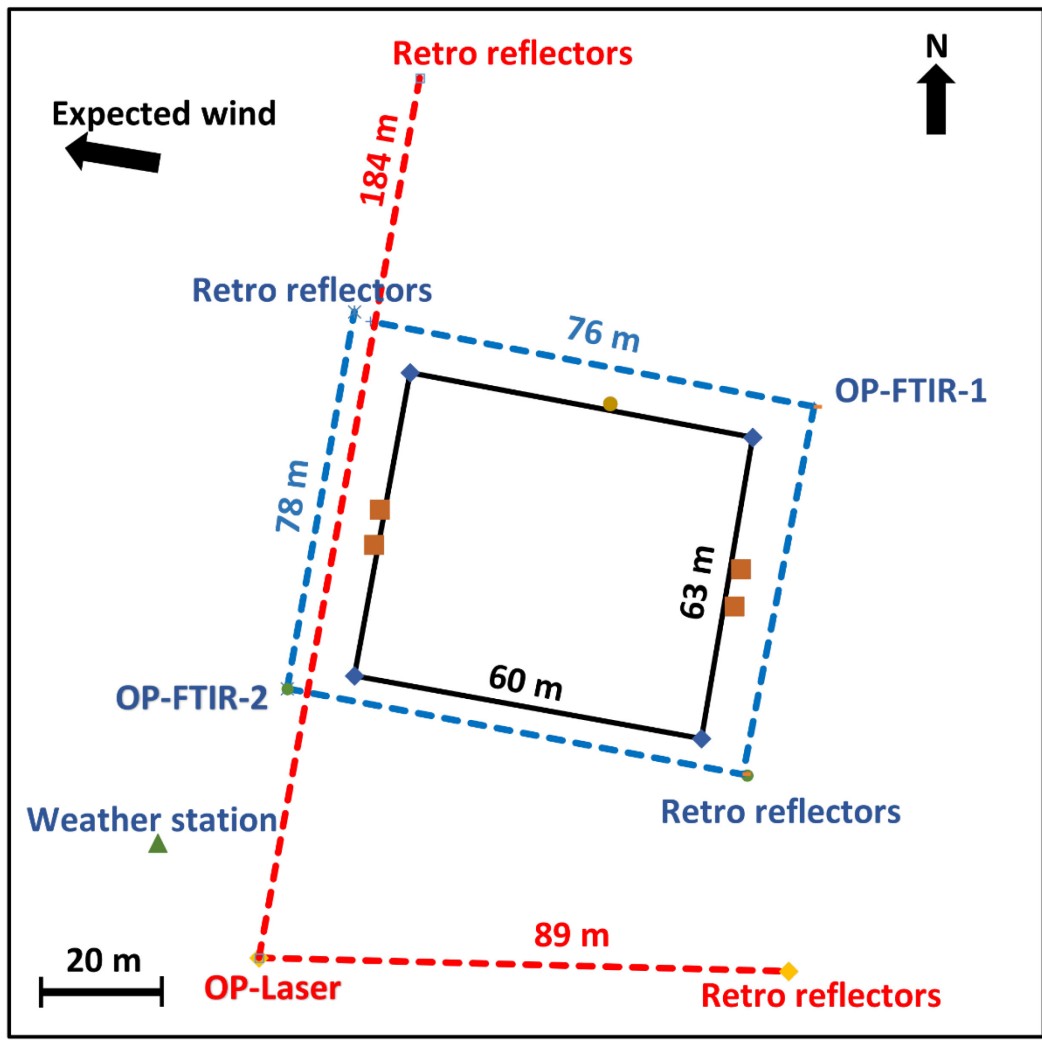

**Figure 1: Schematic layout of the experimental site, showing an animal pen in the center, two OP-FTIR systems (blue dashed lines) and the OP-Laser system (red dashed lines). Two feeding troughs (brown squares) were on both sides of the pen, and one water trough (brown circle) was on the north of the pen. A weather station (green triangle) was 50 m away from the SW corner of the animal pen.**

During the emission measurement period (14 to 21 February 2013) each study animal carried a $N_2O$ canister in a backpack, and controlled rates of $N_2O$ were released as part of the tracer-ratio measurement technique. At 9:00 daily during the measurement period, the 16 study animals were walked from the cattle pen to the adjacent yards (80 m north), and the $N_2O$ gas canister in the backpack was replaced with a fully filled canister. Cattle were absent from the study pen for approximately 15 to 30 min while this occurred. Other than during the canister replacement period, the animals moved and ate freely in the pen while emissions were measured.

**2.2 Concentration sensors**

**2.2.1. OP-FTIR**

Atmospheric concentrations of $CH_4$ and $N_2O$ were measured upwind and downwind of the cattle pen using two open-path Fourier transform infrared (OP-FTIR) spectrometers. OP-FTIR can quantify a wide range of real-time gas concentrations simultaneously with high resolution (Smith et al., 2011). The details of the OP-FTIR system used in this study can be found in Bai (2010) and Paton-Walsh et al. (2014)(2014). Briefly, the modulated infrared (IR) beam from the Bruker IRcube spectrometer (Matrix-M IRcube, Bruker Optics, Ettlingen, Germany) is transferred through the optics to a modified Meade Schmidt-Cassegrain telescope (25.4 cm diameter, Model LX200R, Meade Instrument Corp., Irvine, California, USA) and a secondary mirror, and diverged to 250 mm parallel beam and extended to a distant retro reflector (up to 500 m from the spectrometer) (PLX Industries, Deer Park, New York, USA). The parallel beam is then reflected by the retro reflector and returned to a Mercury Cadmium Telluride (MCT) detector (Infrared Associates Inc., Stuart, Florida, USA) where temperature is controlled by a Stirling cycle mechanical refrigerator cooling system (-196 °C) (Ricor K508, Salem, New Hampshire, USA), as described further in (Bai, 2010). A Zener-diode thermometer (type LM335) and a barometer (PTB110, Vaisala, Helsinki, Finland) provide real-time ambient temperature and pressure data (at the same height of the measurement path) for the analysis of the measured spectra. The spectrometer is operated at 1 cm$^{-1}$ resolution, and one spectrometer scan takes approximately 4 secs (13 scans min$^{-1}$). For acceptable signal to noise ratios, a minimum measurement period of 1 min is required. The measured spectra are quantitatively analyzed using the MALT analysis program and a nonlinear least squares fitting procedure described in Griffith (1996), based on the reference spectra from the molecule absorption databases (HITRAN) (Rothman et al., 2009). The best fitted spectrum is used to retrieve the line-average gas concentrations of $CH_4$ and $N_2O$ over the measurement path. The sensitivity of the OP-FTIR units for $CH_4$ and $N_2O$ is 1 part per billion (ppb), corresponding to 2 and 0.4 ppb for a 100 m path, respectively. To achieve good spectra, parameters including instrument field-of-view (FOV), spectral signal intensity (spec. max), and the residual spectrum between the measured and modelled spectra (RMSresid) are examined. A software "Spectronous" (Ecotech, Knoxfield, Victoria Australia) automatically controls spectrometer, sample collecting, spectrum analysis, data logging and display of the calculated concentrations in real time, together with ambient pressure and temperature.

The OP-FTIR spectrometers were mounted on a motorized aiming system (custom made at the University of Wollongong) to allow the spectrometer to be aimed at different retro reflectors. The two OP-FTIR units were positioned on opposite corners outside the cattle pen, and each unit was alternatively aimed at two reflectors so that gas concentration was measured along the four sides of the pen (Fig. 1). This configuration allowed the downwind $CH_4$ and $N_2O$ enhancements to be measured for any wind direction. The OP-FTIR measurement sequence was repeated automatically so that every 5-min the line-average gas concentration on each path was measured. The average gas concentrations on each of the four paths were averaged over a series of 15-min intervals, from which we calculated a timeseries of $CH_4$ emissions. The OP-FTIR measurement-paths fell approximately 7 m outside the fence line. The distance between the OP-FTIR sensor and retro reflector was either 76 or 78 m, and the measurement path was 1.4 m above the ground.

### 2.2.2. OP-Laser

The open-path laser (OP-Laser) system used a single laser unit (GasFinder2, Boreal Laser Inc., Edmonton, AB, Canada) located outside the animal pen, mounted on a pan-tilt scanning motor (PTU D300, FLIR Motion Control

Systems, Burlingame, CA, USA). The OP laser contains a transceiver that houses the laser diode, drive electronics, detector module and micro-computer subsystems. Collimated light emitted from the transceiver traverses the open measurement path to a distant retro reflector (up to 500 m) and back. A portion of the beam passes through an internal reference cell. The ratio of measured external and reference signals is used to determine the gas concentration from the open path. The retro reflector mounted on a tripod consists of an array of six gold-coated 6 cm corner cubes with effective diameters of approximately 20 cm. The scanning motor was programmed to sequentially measure $CH_4$ concentration on two paths. The paths ran along two sides of the pen, and their location was chosen to provide upwind and downwind concentrations during the prevailing easterly winds (Fig. 1). The two-paths were 89 and 184 m in length, and the laser measurement path was approximately 5 m outside the fence line. The laser alternated between the two paths with a dwell time of 1-min on each path. Line-average $CH_4$ concentration was recorded approximately once a second, and the path average concentrations were averaged into 15-min intervals. The sensitivity of the laser units is 1 part per million-meter (ppm-m), corresponding to 10 ppb for a 100-m path.

### 2.3 Methodologies

A tracer-ratio technique was used to measure $CH_4$ emissions from the study animals. This is a conceptually simple and defensible method for measuring emissions, and we will consider this technique as giving the "true" $CH_4$ emission rate from the animals. Two different implementations of the IDM technique were compared with the tracer-ratio measurements.

### 2.3.1 Tracer-~~ratio~~Ratio technique ~~($N_2O$ Tracer)~~

The tracer-ratio measurements followed the procedure described in Bai (2010), Griffith et al. (2008), and Jones et al. (2011), with $N_2O$ used as the tracer gas and released through a canister at a controlled release rate. The $N_2O$ release point was closed to cattle mouth and nose where the majority of $CH_4$ was emitted. The $N_2O$ tracer gas followed the emitted $CH_4$ downwind of the animal pen, and both concentrations of $N_2O$ and $CH_4$ were measured simultaneously by an OP-FTIR (Fig. 1).

The $N_2O$ tracer (> 99%, BOC Instrument grade, Australia) was released from pressurized canisters (Catalina Cylinders) located in insulated backpacks on each animal. Each canister was fitted with a head encompassing capillary tube (0.025 mm inner diameter, SGE Analytical Science Pty Ltd, Australia) to control the $N_2O$ flow rate. The canister was filled with approximately 300 g of $N_2O$ to provide an average flow rate of 10 g h$^{-1}$ over a 24 h period. The temperature of the canisters was recorded every 5 minutes (Thermochron Temperature model TCS, OnSolutions, Australia). The canisters and temperature sensors were exchanged every 24 h at a nearby yard. Following the procedure in Bai (2010), the canister flow rate was calibrated with a gas temperature dependent factor determined from the measured canister temperature. Canisters were also weighed at the start and end of each 24 h period to get the actual daily $N_2O$ release rate.

The calculation for each pressurized canister $N_2O$ flow rate follows three steps:

1) The $N_2O$ flow rate of each canister was calculated following Bai (2010) (Eq.1):

$Q_{N2O}$ (t) = $Q_0$ + α T (t)               (1)
Where $Q_{N2O}$ (t) is the individual canister flow rate (g h$^{-1}$) at temperature T (°C), t = time, T = temperature °C at
time (t), $Q_0$ is a constant canister flow rate at temperature 0°C, g h$^{-1}$, α is the $N_2O$ flow rate temperature
dependent factor, g h$^{-1}$ °C$^{-1}$. The temperature was measured at 5-min intervals.

2) The integrated $N_2O$ flow rate over the total release time (RT, ~24h) equals the mass loss of $N_2O$ gas ($\Delta m_{N2O}$,
g) (Eq.2):
$Q_0$ = ($\Delta m_{N2O}$ /RT) – (Σ (α T (t)))/RT        (2)
Where $\Delta m_{N2O}$ = $WN_2O_{start}$ -$WN_2O_{end}$
The mass loss of $N_2O$ was determined by the initial and the end weight of the canister (g), $WN_2O_{start}$, $WN_2O_{end}$,
respectively. The integrated $N_2O$ flow rate of each canister was then interpolated to a 15-min interval flow rate
using linear interpolation function (Igor 6.3.7.2). The total $N_2O$ flow rate of the 16 canisters ($Q_{N2O}$) was used for
the $CH_4$ emission rate calculation.

3) Following the procedure described in Bai (2010), Griffith et al. (2008), and Jones et al. (2011), the herd emission
rate of $CH_4$ was calculated (Eq.3):
$Q_{CH4}$ = $Q_{N2O}$*($\Delta CH_4$/$\Delta N_2O$) *($M_{CH4}$/$M_{N2O}$)/$N_{animal}$    (3)
Where $Q_{CH4}$ is the $CH_4$ emission rate, g head$^{-1}$ h$^{-1}$, $Q_{N2O}$ is the integrated $N_2O$ flow rate of total canisters in the
animal backpacks, determined by mass loss of $N_2O$ at canister temperature T and release time t, g h$^{-1}$, is multiplied
by 24 to calculated g head$^{-1}$ d$^{-1}$. The $\Delta CH_4$ and $\Delta N_2O$ parameters are the $CH_4$ and $N_2O$ concentration enhancements
(above the local background level) measured downwind of the animal pen using the OP-FTIR spectrometers,
$M_{CH4}$ is the molecular mass of $CH_4$, 16 g mol$^{-1}$, $M_{N2O}$ is the molecular mass of $N_2O$, 44 g mol$^{-1}$, $N_{animal}$ is ~~animal~~the
number of animals, 16.

During the study we collected a number of air samples using volumetric flasks (600 mL). Samples were spaced
along each measurement path and taken when animals were absent from the pen. These samples were later
analyzed in the laboratory using a closed-path FTIR spectrometer (Griffith, 1996) and the $CH_4$ and $N_2O$ values
were used to cross-calibrate the two OP-FTIR sensors.

Tracer-ratio emission measurements were excluded for periods when the canisters outlets were blocked ~~or~~, had
dropped off the animals, when there was optical misalignment of the OP-FTIRs, or when the enhanced $CH_4$ and
$N_2O$ concentration was less than 50 and 10 ppb, respectively.
**2.3.2 Inverse Dispersion Modelling technique**
Herd $CH_4$ emissions were calculated using the IDM technique (Flesch et al., 2004). This micrometeorological
technique estimates emissions based on the enhancement of $CH_4$ measured downwind of the animal pen. The link
between the concentration enhancement and the pen emission rate is calculated using an atmospheric dispersion
model. The freely available software WindTrax (www.thunderbeachscientific.com) is used for that calculation.
WindTrax combines a backward Lagrangian stochastic dispersion model with mapping software and takes as
input: the upwind and downwind $CH_4$ concentration measurements, wind information from a sonic anemometer,

and a map of the pen and gas sensor locations. General information on WindTrax applications is given in Flesch and Wilson (2005).

The upwind and downwind $CH_4$ concentration was measured using either the OP-FTIR system previously described (designated IDM-FTIR) or by an open-path $CH_4$ laser system (designated IDM-Laser). Air samples collected during the study were used to cross-calibrate the laser and the OP-FTIR sensors (applying a retroactive correction multiplier to the laser concentrations). Air samples were collected at 2-min intervals to get 15-min average concentrations for the period from 9:15 to 9:30 when the cattle were not in the paddock. The samples were analyzed using aby gas chromatographchromatography (Agilent 7890) at the University of Melbourne laboratory. Three positions were sampled: 1) directly west of the paddock along the laser/FTIR line, 2) near the laser, southwest of the paddock, and 3) far south of the paddock along the southerly laser line. Winds were from light and from the east. We assumed the $CH_4$ and $N_2O$ concentrations at these positions would be similar (as cattle were absent) and would provide the basis for calibration of the lasers and FTIRs.

A weather station southwest of the cattle pen (Fig. 1) included a 3-dimensional sonic anemometer (CSAT-3, Campbell Scientific Inc, Logan Utah, USA) mounted 2.45 m above the ground. The anemometer provided the wind information needed for the IDM calculation, including the friction velocity ($u_*$), Obukhov stability length ($L$), average windspeed and wind direction, and the standard deviation of the velocity fluctuations in the three directional components ($\sigma_{u,v,w}$). The surface roughness length ($z_0$) was calculated from these variables (Garratt, 1992). The wind variables were averaged into 15-min intervals matched to the gas concentration dataset.

### 2.3.3 Data filtering criteria

The $CH_4$ emissions were calculated in 15-min intervals using the WindTrax software. We defined the $CH_4$ as coming from an elevated area source 0.8 m above ground, which overlaid the pen area. In the IDM analysis we followed the procedure of Flesch et al. (2005) to remove error-prone intervals when either $u_* < 0.15$ m s$^{-1}$, $|L| < 5$ m, $z_0 < 0.9$ m, or when the fraction of WindTrax trajectory touchdowns inside the pen source covered $< 10\%$ of the pen area. Intervals were also removed when the concentrations measured by the OP-FTIR or the laser corresponded to low signal levels: i.e., FOV < 35, RMSresid < 0.2%, spec.max was < 0.25 in the spectral region of 2200 cm$^{-1}$ for the OP-FTIR, or the light level reported by the laser fell outside the 2000 to 13000 range, or the laser quality parameter $R^2 < 0.97$.

### 2.3.4 Calculating Average Emissionsaverage emissions

The tracer-ratio and IDM measurements are a discontinuous time series of 15-min average emission rates lasting for seven days. In order to create a properly weighted daily average emission rate, these discontinuous data were used to create an ensemble 24-h diel emission "curve" for each technique. Each emission observation was binned into one of the 96 15-min periods making up the ensemble day. We used Generalized Additive Models (GAM) fitted to the time series of gas emission to impute missing measurements (Bai et al., 2020). The time series of gas emission and associated GAM fit for each measurement method are shown in Appendices (Fig. A1). The average daily emission rate was calculated by summing the 15-min emission intervals over the 24 h day.

Following IPCC (2006) ~~recommendation~~recommendations, CH$_4$ ~~emission using~~emissions were also calculated
based on DMI (Eq. 10.21). This assumes CH$_4$ energy content = 55.65 MJ (kg CH$_4$)$^{-1}$, DMI energy content = 18.45
MJ (kg DMI)$^{-1}$, and CH$_4$ conversion factor Y$_m$ = 6.5%.

## 3 Results

### 3.1 Climate condition

During the seven-day emission measurement period the total rainfall was 0.4 mm, the average minimum and
maximum ambient temperature was 12.9 and 22.4 °C, respectively. The wind speeds (at 2.45 m above ground)
varied from 2 to 8 m s$^{-1}$, and the wind direction was predominately from the east (Fig. 2). This period had excellent
conditions for the micrometeorological measurements due to the lack of precipitation, the absence of light wind
periods, and the steady easterly winds.

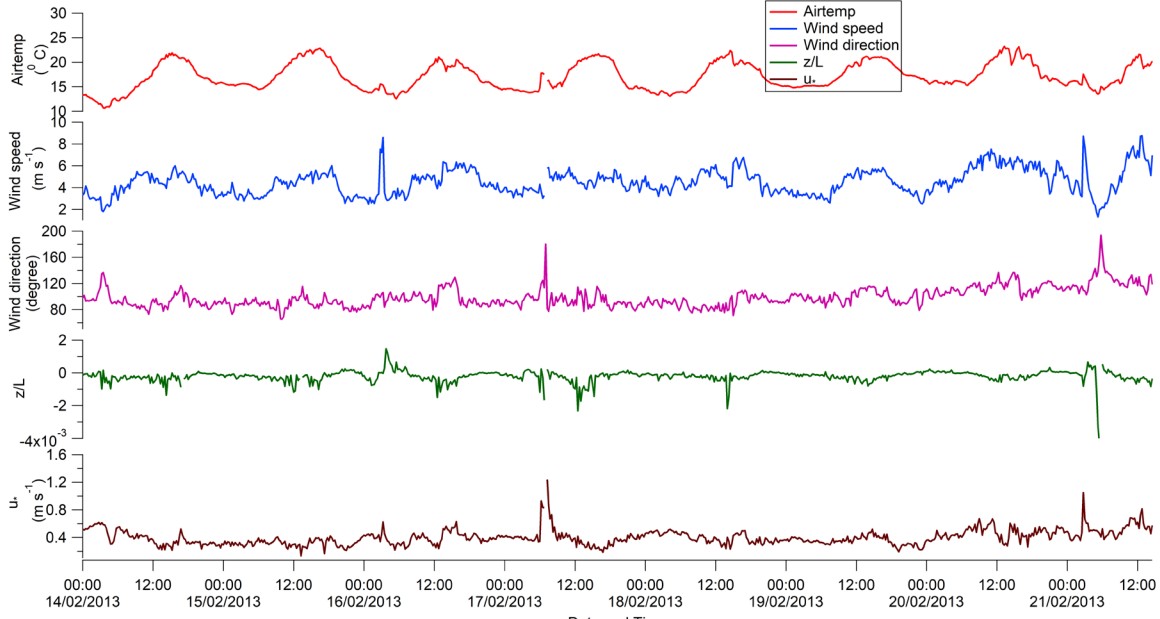


**Figure 2 Ambient temperature (Airtemp), wind speed, wind direction ~~was~~ measured during the study. Atmospheric**
**stability parameter (z/L) and wind friction velocity (u$_*$) are also plotted.**

### 3.2 Methane emission measurements

#### 3.2.1 Tracer-~~ratio~~Ratio measurements

The OP-FTIR system measured downwind CH$_4$ enhancements between 50 and 150 ppb, and N$_2$O enhancements
between 12 and 30 ppb over the study (Fig. 3). These enhancements are well above the minimum sensitivity of
the OP-FTIR given by Bai (2010) of 2 ppb for CH$_4$ and < 0.4 ppb for N$_2$O. Over the seven study days, emissions
were measured during 90% of the ensemble 24 h day (i.e., 86 of the 96 possible 15-min periods). The average
daily emission rate (± standard error) from the tracer-ratio technique was 301.9 (± 1.5) g CH$_4$ head$^{-1}$ d$^{-1}$ (Table 1).



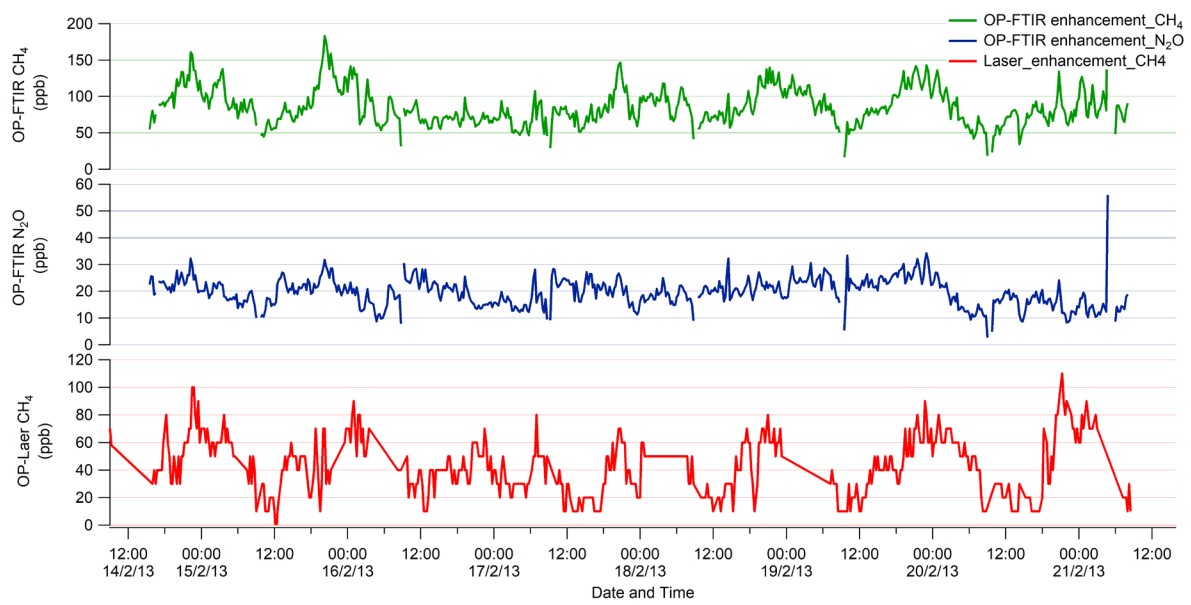

Figure 3. The concentration enhancement of N₂O and CH₄ from OP-FTIR and CH₄ from OP-Laser over the measurement period of 14−21 February 2013.

Table 1. Methane emission rates from the three micrometeorological measurements (Tracer-Ratio, IDM-FTIR, IDM-Laser) and an emission estimate based on the dry matter intake of the animals (using an IPCC recommended calculation[§]). Methane yield (g CH₄ kg⁻¹ DMI) is also shown.

| | Emission Rate (g CH₄ head⁻¹ d⁻¹) | Methane Yield (g CH₄ kg⁻¹ DMI) |
|---|---|---|
| Tracer-Ratio | 301.9 (1.5) | 27.0 |
| IDM-FTIR | 308.1 (2.1) | 27.0 |
| IDM-Laser | 304.4 (8.0) | 27.1 |
| IPCC[§] | 254[§] | 21.3 |

[§]IPCC (2006) calculation based on DMI (Eq. 10.21). Assumes CH₄ energy content = 55.65 MJ (kg CH₄)⁻¹, DMI energy content = 18.45 MJ (kg DMI)⁻¹, and CH₄ conversion factor $Y_m$ = 6.5%.

### 3.2.2 ~~The inverse-dispersion modelling (IDM) emissions~~3.2.2 Inverse Dispersion Modelling measurements

Over the seven-day study, 90% of the ensemble was represented with the IDM-FTIR measurements, and 79% was represented by the IDM-Laser measurements. The majority of missing periods resulted from instrumental issues (e.g., low signals caused by condensation on mirrors, power failure), and to a lesser extent by inappropriate meteorological conditions (e.g., low wind speed, $u_* < 0.15$ m s⁻¹). The 24-h diel CH₄ flux over the measurement period is shown in Figure 4. There are differences between the three ensemble emission relationships in Figure 4. We assume the tracer-FTIR data is most accurate data set. Differences between the tracer and IDM approaches are due to a combination of a less-sensitive laser sensor (compared to the OP-FTIR) and the incorrect assumption that animals were spread evenly over the pen (which effects the FTIR and laser estimates differently due to different measurement locations). Both of the IDM-FTIR and tracer-ratio measurements show a similar emission pattern: emission rates at a minimum around 9:00 local time, and at a maximum during the early evening. This

emission peak pattern reflected the time when animals were fed, or the pellets were topped up. However, IDM-
Laser shows a late minimum emission at 12:00 local time, likely due to a solar related alignment of the retro
reflector. We calculated average daily emission rates of 308.1 (± 2.1) and 304.4 (± 8.0) g $CH_4$ head$^{-1}$ d$^{-1}$ for the
IDM-FTIR and IDM-Laser measurements, respectively (Table 1). These results are not statistically different from
each other. Both IDM estimates were not statistical different from the tracer-ratio results.

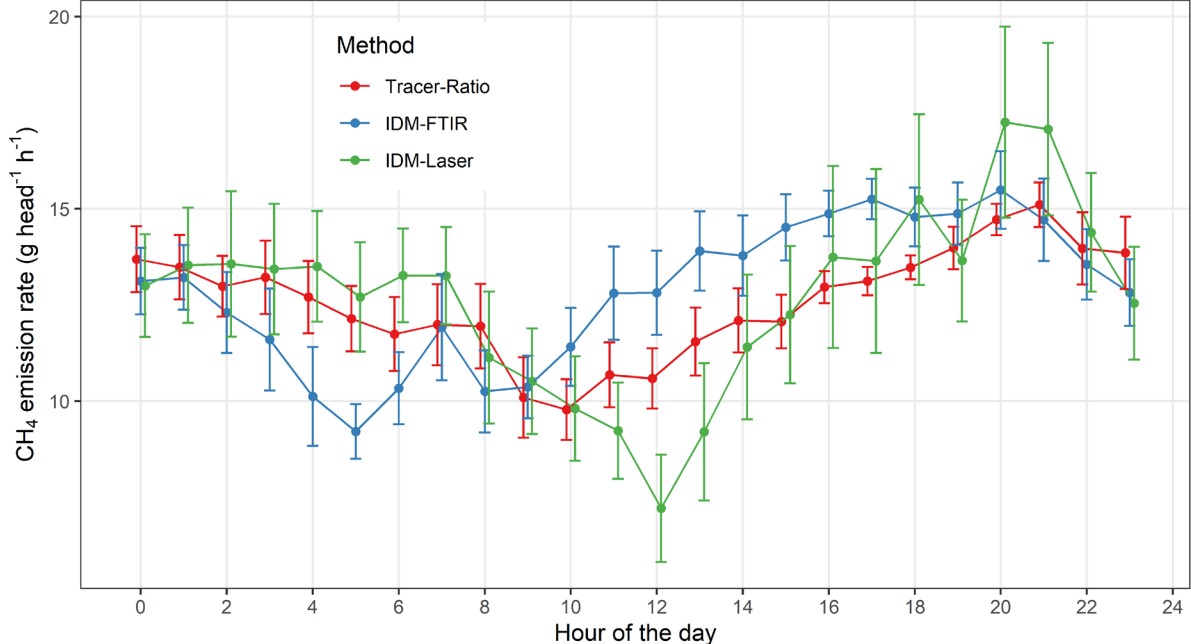


**Figure 4: Ensemble 24-h diel $CH_4$ emission pattern measured by IDM-Laser, IDM-FTIR, and Tracer-Ratio method**
**(hourly values based on 7-d of measurements). Error bars denote the standard error of mean.**

**4 Discussion**
There was excellent agreement between the tracer-ratio and the IDM measurements of cattle $CH_4$ emissions (there
were no statistical differences between the different techniques). For potential users of the IDM technique, these
results are an important finding. When previously applied to cattle environments, some recent IDM studies have
monitored animal positions assuming this information is critical to getting accurate calculations (e.g., McGinn et
al., (2011)). Alternatively, other studies constrained animal locations by fencing to minimize the errors when
animal positions were not monitored (Flesch et al., 2016). However, our IDM calculations assuming cattle were
evenly distributed across the paddock were nearly identical to the tracer-ratio results that implicitly include the
impact of animal positions. This indicates that IDM studies like ours can use the much simpler approach where
the whole paddock is treated as a gas source, and animal positions need not be monitored. This seems to confirm
a similar finding from McGinn et al. (2015). The effect of this simplification on measurement accuracy is likely
to depend on animal density and the size of the paddock. For example, the measurement of a small number of
animals in a large paddock is likely to be very sensitive to the exact animal positions. But in the modest sized
paddock studied here (and in McGinn et al., (2015)) this is not the case.

It is interesting to compare our measured $CH_4$ emission rates with estimates made using the IPCC (2006) suggested relationships based on DMI. Using the IPCC recommendations that $CH_4$ emissions represent 6.5 % of the gross energy intake of the cattle ($Y_m$) and with our DMI = 11.9 kg $d^{-1}$, we calculate (Eq. 10.21) an emission rate of 254 g $CH_4$ $head^{-1}$ $d^{-1}$. Using the equation from Charmley et al. (2016) ~~and~~ with the yield of 20.7 g $CH_4$ $kg^{-1}$ DMI, the estimated $CH_4$ ~~emissions~~emission rate is 246 g $CH_4$ $head^{-1}$ $d^{-1}$. The DMI based $CH_4$ estimates were lower than the tracer-ratio measurement of 321 g $CH_4$ $head^{-1}$ $d^{-1}$. What might explain this difference?

- Weather conditions during our study were nearly ideal for the micrometeorological calculations, resulting in a large and representative set of emission calculations over the study, and a good estimate of the 24-h ensemble daily emission rate. A time-of-day sampling bias in the tracer-ratio measurements is unlikely to cause the difference.

- Differences between the tracer-ratio and IPCC estimated rates would occur if there were significant manure or rectal emissions that are measured by the micrometeorological techniques, but not reflected in the IPCC estimates. However, the general view is that these emissions are small in comparison to enteric emissions (Flessa et al., 1996; Kebreab et al., 2006; McGinn et al., 2019). In addition, when animals were absent from the pen, we did not observe enhanced $CH_4$ levels downwind of the pen, indicating low emission rates from the pen manure. There were no manure stockpiles nearby during the study. This suggests that IPCC estimates may have larger uncertainties.

- Based on the tracer-ratio measurements, the $CH_4$ conversion factor $Y_m$ in this study is higher than the IPCC suggested value, that is: our measured $Y_m$ of 8.3 % is outside the 6.5 ± 1 % range suggested by IPCC (2006). However, the IPCC suggestion is a rough estimate, and several grazing studies have found $Y_m$ values higher than our 8.3 % (e.g., Tompkins and Charmley (2015); McGinn et al. (2011); Ominski et al. (2006)).

## 5 Conclusions

We are very confident in the tracer-ratio measurements given the conceptual simplicity of the approach (where each animal is a tracer gas source), given that the OP-FTIR is a very sensitive gas sensor, and given the agreement between the associated IDM measurements. We thus view the relatively high emission rates we observed to be representative of the conditions of the study.

The (external) tracer ratio technique is a "gold standard" for measuring cattle emissions in an ambient outdoor environment. However, this technique is difficult to use given the need to outfit the animals with tracer sources, and to monitor tracer gas concentrations downwind. Encouragingly, our results indicate that a logistically simple IDM technique can provide an accurate tool for measuring emissions from cattle, with far greater practicality than the tracer-ratio technique. It is worth noting that micrometeorological methods like IDM represent one of the major approaches for measuring cattle emissions (in addition to internal $SF_6$ tracer technique and respiration chambers). Our results should give users added confidence that a practical micrometeorological technique can provide an accurate method of estimating $CH_4$ emissions at farm scales.

## 6 Appendices

Appendix A

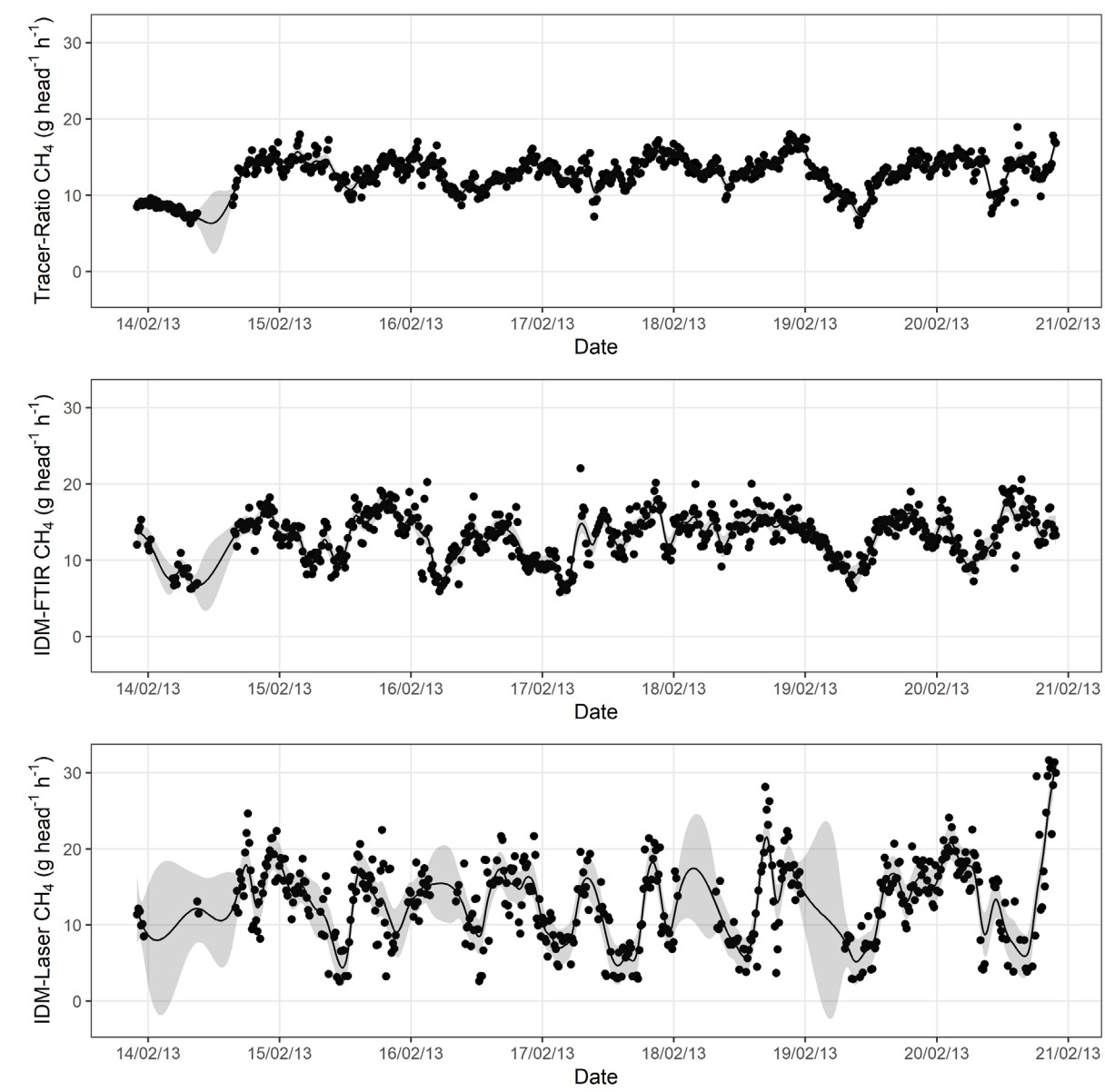

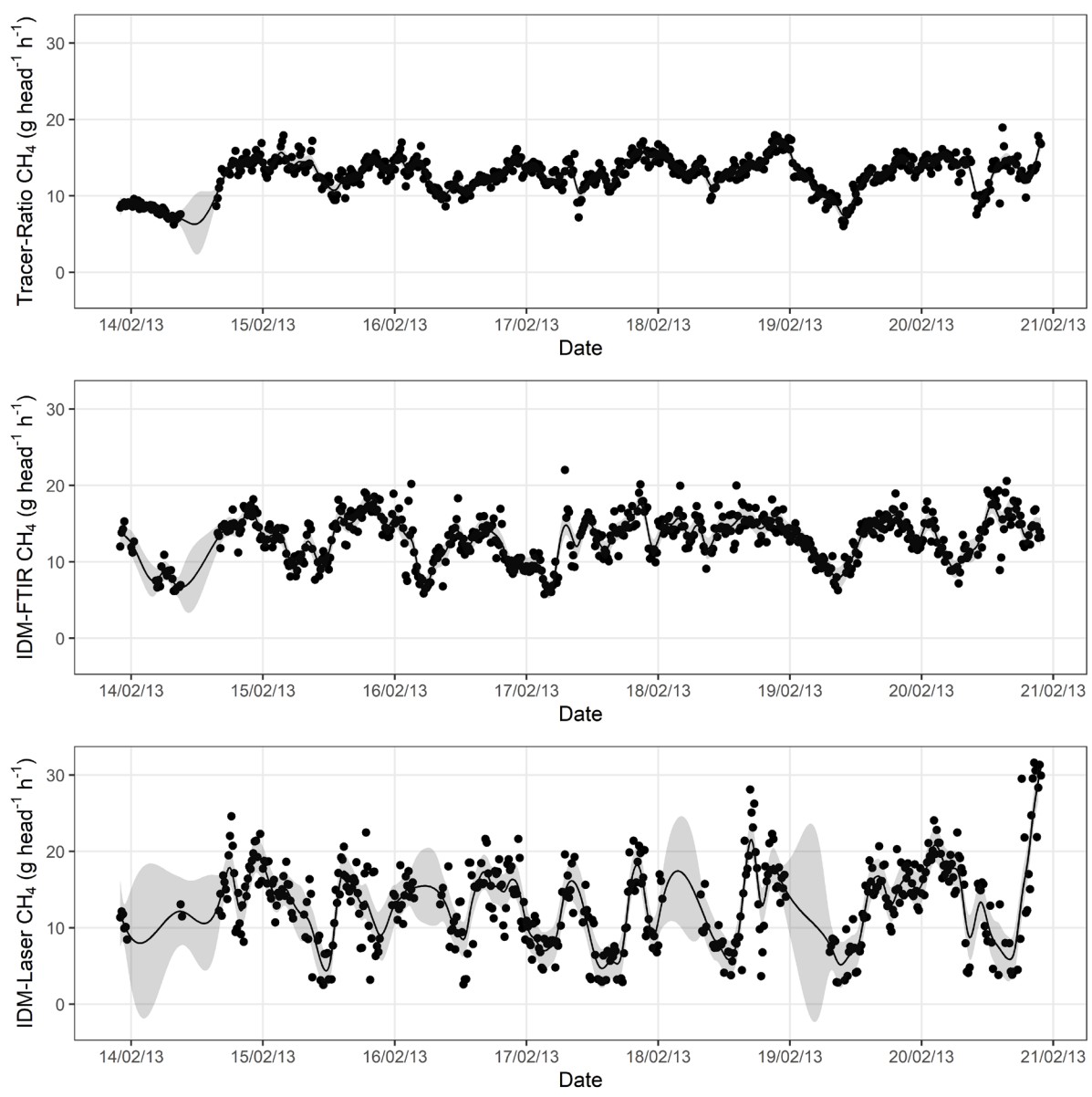

**Figure A1: Time series of CH₄ emissions measured using the Tracer-Ratio, IDM-FTIR, and IDM-Laser methods. Black dots show the 15 minutes measurements. The solid black line shows the mean value of gas emission estimated from a GAM fit to the measurement data. The shaded area represents the 95% credible intervals of the mean gas emission from the GAM fit (i.e., it contains 95% of the potential mean values of gas emission at a given time).**

## 7 Data availability

The raw data are not available to the public. For any inquiry about the data, please contact the corresponding author (mei.bai@unimelb.edu.au).

## 8 Author contributions

All authors contributed to the conceptualization, methodology, draft writing, and original draft preparation. TC, JIV, TF, FP, MB, and NT contributed to writing, reviewing and editing. TF, JIV, TC, FP, and MB contributed to formal analysis. DC, RH, NT, JH, DM contributed to funding acquisition and investigation.

## 9 Acknowledgements

This study was funded by the CSIRO Sustainable Agriculture Flagship Program. José I. Velazco was supported
by an Australian Government scholarship funded by the Australian Agency for International Development and by
the National Institute for Agricultural Research (INIA Uruguay). We thank Travis Naylor, Kithsiri Dassanayake,
and Jianlei Sun for their field support. We also thank Sheilah Nolan for editing the manuscript and Raphaël Trouvé
for helping with the GAMs.
**10 Declaration of interests**
The authors declare that they have no known competing financial interests or personal relationships that could
have appeared to influence the work reported in this paper.

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
