# Peer review of "Beef cattle methane emissions measured with tracer-ratio and inverse dispersion modelling techniques"

_Atmospheric Measurement Techniques, 2020_

## Referee Comment (RC1) · Anonymous Referee #1 · 21 Dec 2020

The Authors submitted a manuscript outlining the measurement of methane emissions from an experimental pen with beef-cattle in Australia. The work presents continuous measurement for 7 days in a row and compare two different methods: trace-ratio and inverse dispersion modelling (IDM). The IDM method, in particular, is presented with two different concentration measurement approaches, a Laser, and an open-path FTIR. The Authors presented an interesting experimental set-up and compared the methods, stating that the IDM is accurate to perform the measurement in the proposed set-up and meteorological conditions. The methodology presented is lacking and needs to be reorganised and completed to improve the understanding of the work. The results do not show sufficient elements to meet the objectives of the paper and the discussion

part must be strengthened. The paper concluded that IDM is a suitable methodology to perform this type of measurements, grounding this statement by comparing only the cumulative emissions from different measurement techniques. No micrometeorological parameter have been detailed and discussed and no prove of the validity of the application of the IDM method was provided (e.g. homogeneity of the source). On the other hand, the experimental set-up and the dataset of the paper have the potential to contribute to this journal, in a decidedly revised form. For these considerations, I suggest a strong revision of the manuscript before the publication in this journal, which takes into consideration the comments below. I therefore recommend major revisions, with the evaluation of the revised paper.

** Abstract **

1. Should be reconsidered based on the remark of the other sections. I would suggest to add the details of the experiment's management, e.g. duration of the trial, pointing that was an experimental pen. Adding the type of the FTIR and the Laser used can be useful to understand the method used. I would suggest also to add the uncertainty of the cumulative flux.

** Introduction **

2. The introduction is somewhat lacking and should be extended. Moreover, some elements missing to understand the novelty of this work compared to literature. First, I suggest adding some data from the inventories related to the contribution of the livestock to GHG emissions in Australia and the World. This Reviewer suggest a series of questions to be answered with the purpose to improve the introduction section: What is $CH_4$ and how it affects the climate? Where does $CH_4$ emissions come from? How agriculture (and livestock) contributes to $CH_4$? Which are the most prominent mitigation options?

3. Secondly, the section describing the different available methods and techniques to measure enteric $CH_4$ is, in my opinion, poor and should be improved. I suggest to
add examples from literature, which can be used also in the discussion section (this Reviewer add here a non-exhaustive list: Felber et al., 2015; Dengel et al., 2011; Lockyer and Jarvis, 1995; Grainger et al., 2007; Laubach et al., 2008; Todd et al., 2014). The strengths and weaknesses of each measurement method or technique should be stressed in order to defend the type of methodologies used by this study. I suggest also adding some experience about the tracer-ratio technique, since it is defined as "true" in this paper (L 69).

4. Third, the novelty of this study. If the IDM technique has been already applied to perform the quantification of CH4 from grazing animals, I would encourage adding these information and stressing how the work you are presenting has some novelty (technical, methodological, or environmental conditions) compared to the literature and previous studies (i.e. Bai 2010). Finally, why do you compared two different concentration measurement tools?

** Materials and Methods **

5. This section should be reorganised. I suggest providing a detailed section of the laser and FTIR, along with their working principle and field setup, just below the Experimental Design section, then describe the two methods (tracer-ratio and IDM). The information about the FTIR and Laser is scattered and not well organised, disadvantaging readability and understanding. See also the comments #10 to 12; and #22. I suggest including here the details of the calculation of the losses made with IPPC's guidelines (Table 1); comment #21

* Experimental design: *

6. I suggest adding (here or in the Results section, see comment #19) more details about the experimental site, as the meteorological variables measurements from the weather station (e.g. rain, temperatures, wind direction and speed). This will help the reader to understand the validity of the measurements (e.g. wind direction), this particular environment and, of course, the results.

7. Can the Authors detail here more about the dejections management during the experiment in order to better understand the field set-up and neglecting further sources of methane?

8. L 54-55 I would suggest to add a reference to the part of the tracer description (2.2.1)

9. Figure 1. I warmly suggest re-making the picture with a proper scale. This will help better understating the field setting and the distances of the probes (and the weather station) from the fences.

* Methodologies * * Tracer-ratio technique (N2O Tracer) *

10. I would suggest to explain with more detail and in a few line how is the principle of this method. L75, please explain what is QCH4 in the text.

11. Can the Authors add the details of the producer of the FTDIR, the measurement range (to justify also lines L112-113 and L142-145), the uncertainty and sensitivity (to justify lines L162, L178), and all the technical parameters that can help to characterise this measurement. Could the Authors detail where the measurements were recorded ?

12. Can the Authors add some information about the close-path FTIR used in laboratory in this section, or in the section of the concentration measurements (see comment #5)?

* Inverse Dispersion Modelling technique *

13. This section should represents one of the main methodological part of the paper and, I retain, it can be improved. I suggest adding the principle behind the backward application of the short-range Lagrangian dispersion model used in the study (equation, number of trajectories used and principles of the MOST). This will improve the understanding of the scientific ground, the application of this technique in the case study presented here and better understand the Equation 1. Any reference to other

study using IDM in the short range is recommended.

14. Please, detail how the roughness length was calculated (reference or equation, and the results). Furthermore, can the Authors detail if they're using a constant or a variable z0.

15. L139-142. Can you address why these thresholds were imposed for this case study and why these are different from Flesch et al 2005? Please, refer here to the methodological part requested in the comment #13. Furthermore, how many "15-mins" data were excluded from the dataset with these thresholds and in which part of the day?

16. L 143. Please, explain what "spec.max" stands for.

*Calculating Average Emissions*

17. I would suggest to rewrite this part more clearly, giving some reference to other studies which use the same calculation. This will greatly help the reader. A gap-filling procedure has been used? Pleas add these details.

18. Were the periods when the animals were not in the pen excluded from the measurement dataset? This point should be better described.

** Results **

*Climate condition*

19. This part should be improved and extended. I would suggest adding a figure with the dynamics of air temperature, wind speed and rain, at least. Moreover, I warmly recommend to add a figure with the trends of u* and the turbulence parameter z/L.

*Tracer-ratio measurements*

20. To better understand the measurement performed, given that two different methods are compared in this study (Laser and FTIR), it might be interesting to evaluate the concentrations observed over time by the two systems and by the tracer, before

evaluating the final daily cumulative emissions. I suggest to provide these results.

21. Table 1. I suggest putting the measurement uncertainty for each of the measurements. I would also suggest removing the reference (Charmley) from the table and keep it exclusively in discussions section along with the other sources cited to defend your findings. Furthermore, I would better explain the calculation with the IPCC's guidelines in materials and methods (see comment #5).

22. L161-163. What about the sensitivity of the laser source?

* The inverse-dispersion modelling (IDM) emissions *

23. L175. I would suggest to detail better what "low wind speed" means for the Authors. Or, if these percentages are comprehensive of the periods not considered because of the MOST conditions failure (L139-142)?

24. I cannot see any comparison about the "sensitivity" of the two sensors. I suggest to address this part on the Materials and Methods section (see comment #5) and in the results (comment #19).

25. The lowest emission value is at 9 am, the time when the animals left the pen. How did this event affect the dataset? Are these gaps filled and how ?.

26. I would warmly suggest to insert a further figure about the trend of 15-mins emissions over the 7 days of measurement. This will give the real picture of the dataset, without the period of failures (technical), filtered because of the MOST failure.

27. Figure 2. IDM-FTIR does not have the measurement at 11pm.

** Discussion and conclusion **

28. The discussions should be better set up and expanded with other literature studies to defend the validity of the measurement, i.e. defending that the conditions of the experiment were always suitable for the application of the IDM. It seems that the reliability of the IDM method is related only to the final cumulative emissions (Table 1). In order to

define that the source was homogeneous, and therefore the monitoring of the animals is not needed, as stated, further results from this study - or results from other studies - should be provided.

29. Referring only to the method of the IPCC guidelines is, in my opinion, limited. I would suggest broadening the discussions with other case studies, reporting their characteristics and results to make the measurement more robust (e.g. references cited online 226).

30. The conclusions, with respect to the use of IDMs, should be much more cautious given that this is an experiment of only 7 days, performed in micrometeorological conditions not detailed in the paper, without a real defence of the validity of the application of the method itself (homogeneity of the source).

** References cited in this document **

Felber, R., A. Muenger, A. Neftel, and C. Ammann. 2015. Eddy covariance methane flux measurements over a grazed pasture: Effect of cows as moving point sources. Biogeosciences 12:3925–3940. doi:10.5194/bg-12-3925-2015

Dengel, S., P.E. Levy, J. Grace, S.K. Jones, and U.M. Skiba. 2011. Methane emissions from sheep pasture, measured with an open-path eddy covariance system. Glob. Change Biol. 17:3524–3533. doi:10.1111/j.1365-2486.2011.02466.x

Laubach, J., F.M. Kelliher, T.W. Knight, H. Clark, G. Molano, and A. Cavanagh. 2008. Methane emissions from beef cattle- a comparison of paddock-and animal-scale measurements. Aust. J. Exp. Agric. 48:132–137. doi:10.1071/ EA07256

Lockyer, D., and S. Jarvis. 1995. The measurement of methane losses from grazing animals. Environ. Pollut. 90:383–390. doi:10.1016/0269-7491(95)00009-G

Todd, R.W., M.B. Altman, N.A. Cole, and H.M. Waldrip. 2014. Methane emissions from a beef cattle feedyard during winter and summer on the southern high plains of Texas. J. Environ. Qual. 43:1125–1130. doi:10.2134/ jeq2013.09.0386

Grainger, C., T. Clarke, S.M. McGinn, M.J. Auldist, K.A. Beauchemin, M.C. Hannah, et al. 2007. Methane emissions from dairy cows measured using the sulfur hexafluoride (SF6 ) tracer and chamber techniques. J. Dairy Sci. 90:2755–2766. doi:10.3168/jds.2006-697

---

## Referee Comment (RC2) · Anonymous Referee #2 · 19 Feb 2021

The authors describe results from a measurement campaign, measuring methane emissions from cattle, comparing two techniques (based on either a tracer gas or inverse dispersion modelling). They find similar results from both techniques, which give higher estimates than the IPCC default. The paper is generally well-written and nicely succinct. However, it requires some more detail in a few areas, and there is considerable room for improvement.

General points:

1. Two different laser instruments were used, as well as the comparison of two different measurement techniques. The rationale for this is not very well explained, and needs

expanded upon.

2. The IDM method is barely explained, and really only by reference to previous papers and the software web site. This needs at least a brief description and some key details.

3. How were the uncertainties calculated? This is important, and values needs to be included in Table 1. Statistical testing is not relevant, and reference to p values should be removed.

4. The advantage of the IDM method versus the tracer method is not very clear. The latter is simple enough to be explained in one equation. It seems to be the logistic problem of releasing the tracer versus the computational task and assumptions of running WindTrax. Scope for discussion at least.

5. Discussion - the agreement of the two method seems to depend on the spatial distribution of the animals, which comes down to the vagaries of the pen size and animal density. So a single 7-day experiment is not grounds to say that the agreement will generally be good. Does the spatial distribution of the animals affect both methods similarly? If not, why not? This needs expanding.

Specific points:

l 72: the tracer-ratio technique is indeed simple, but cannot be considered "true". The uncertainty associated with its estimates needs to be quantified. The point is presumably that these uncertainties are smaller than the IDM method, but this needs to be demonstrated. E.g. how predictable is the N2O release rate? The authors say this has to be corrected for temperature dependency, but presumably this is established in lab tests?

l 78: define exactly what $Q_{ch4}$ is, with units.

l 110: could the data collected while the animals were absent be shown, to demonstrate the noise/sensitivity? This provides a neat control period with zero emission.
l 131: how big is the sensor drift? Is this a large uncertainty?

l 144: touchdowns - the whole Lagrangian particle idea needs to be explained.

l 146: spec.max?

l 149: why was the diel cycle used? What is driving the diel cycle in methane emission? Production rate should be constant, but emission will be affected by feeding behaviour, Or is this cycle a measurement artefact? This needs explaining. Other gap-filling methods might be better e.g. smoothers such as GAMs.

l 171: Yield needs to be explained.

l 216: This question is not answered very clearly. The authors seem to find no reason, except the IPCC value should perhaps have wider uncertainty bounds on it.

---

## Author Comment (AC2) · 18 Mar 2021

Dear reviewer (Anonymous Referee #2), Re: Revision of manuscript Number: AMT-2020-445, Title: Beef cattle methane emissions measured with tracer-ratio and inverse dispersion modelling techniques

We thank your positive feedback on the manuscript. We have addressed the comments thoroughly, our response to every issue raised is given point by point below.

1. Two different laser instruments were used, as well as the comparison of two different measurement techniques. The rationale for this is not very well explained, and

needs expanded upon. Agree. In the revised manuscript, the information has been added in the Introduction section as well as the methods and materials. 2. The IDM method is barely explained, and really only by reference to previous papers and the software web site. This needs at least a brief description and some key details. Agree. we have added the information in the revised manuscript. Pages 6-7, lines 203-210. "Herd CH4 emissions were calculated using the IDM technique (Flesch et al., 2004). This micrometeorological technique estimates emissions based on the enhancement of CH4 measured downwind of the animal pen. The link between the concentration enhancement and the pen emission rate is calculated using an atmospheric dispersion model. The freely available software WindTrax (www.thunderbeachscientific.com) is used for that calculation. WindTrax combines a backward Lagrangian stochastic dispersion model with mapping software and takes as input: the upwind and downwind CH4 concentration measurements, wind information from a sonic anemometer, and a map of the pen and gas sensor locations. General information on WindTrax applications is given in Flesch and Wilson (2005)." 3. How were the uncertainties calculated? This is important, and values needs to be included in Table 1. Statistical testing is not relevant, and reference to p values should be removed. Agree. We have added the standard error of each mean values. The standard errors have been added in the Table 1. Also removed the p value.

4. The advantage of the IDM method versus the tracer method is not very clear. The latter is simple enough to be explained in one equation. It seems to be the logistic problem of releasing the tracer versus the computational task and assumptions of running WindTrax. Scope for discussion at least. We agree with the reviewer's comments. We have added the information of tracer-ratio and IDM method to the revised manuscript. Page 2, lines 38-51.

5. Discussion - the agreement of the two method seems to depend on the spatial distribution of the animals, which comes down to the vagaries of the pen size and animal density. So a single 7-day experiment is not grounds to say that the agreement

will generally be good. Does the spatial distribution of the animals affect both methods similarly? If not, why not? This needs expanding. In the manuscript, we have explained the good agreement between the IDM and tracer-ration technique. Page 10, lines 306-316. "When previously applied to cattle environments, some recent IDM studies have monitored animal positions assuming this information is critical to getting accurate calculations (e.g., McGinn et al., (2011)). Alternatively, other studies constrained animal locations by fencing to minimize the errors when animal positions were not monitored (Flesch et al., 2016). However, our IDM calculations assuming cattle were evenly distributed across the paddock were nearly identical to the tracer-ratio results that implicitly include the impact of animal positions. This indicates that IDM studies like ours can use the much simpler approach where the whole paddock is treated as a gas source, and animal positions need not be monitored. This seems to confirm a similar finding from McGinn et al. (2015). The effect of this simplification on measurement accuracy is likely to depend on animal density and the size of the paddock. For example, the measurement of a small number of animals in a large paddock is likely to be very sensitive to the exact animal positions. But in the modest sized paddock studied here (and in McGinn et al., (2015)) this is not the case." Given our experiment configuration, constant easterly winds, absence of light winds, and good quality data that 80-90% of measurement data is useful and sufficient to calculate the fluxes, we are confident with our results based on 7-day measurement.

Specific points: I 72: the tracer-ratio technique is indeed simple, but cannot be considered "true". The uncertainty associated with its estimates needs to be quantified. The point is presumably that these uncertainties are smaller than the IDM method, but this needs to be demonstrated. E.g. how predictable is the N2O release rate? The authors say this has to be corrected for temperature dependency, but presumably this is established in lab tests?

The canister pressure is correlated with the ambient temperature. Prior to the field study, temperature dependent factor associated with canister release rate was tested

СЗ

in the laboratory, and CO2 was used as the tracer gas due to its low cost. CO2 flow rate was tested at a range of controlled temperature to determine the CO2 release rate temperature dependent factor. Because N2O and CO2 have the same molecular mass, we assumed that N2O and CO2 disperse in a similar path in the atmosphere and have the same characteristic temperature dependence. The N2O flow rate calculation consists of three steps:

1) The N2O flow rate of each canister was calculated following Bai (2010) (Eq.1): QN2O(t) = Q0 + EST(t) (1) Where QN2O(t) is the individual canister flow rate (g h-1) at temperature T (°C), t = time, T = temperature °C at time (t), Q0 is a constant canister flow rate at temperature 0°C, g h-1, ÉŚ is the N2O flow rate temperature dependent factor, g h-1 °C-1. The temperature was measured at 5-min intervals. 2) The integrated N2O flow rate over the total release time (RT,  $\sim$ 24h) equals the mass loss of N2O gas ( $\Delta$ mN2O, g) (Eq.2): Q0 = ( $\Delta$ mN2O /RT) – ( $\Sigma$  (ÉŚ T (t)))/RT (2) Where  $\Delta$ mN2O = WN2Ostart -WN2Oend The mass loss of N2O was determined by the initial and the end weight of the canister (g), WN2Ostart, WN2Oend, respectively. The integrated N2O flow rate of each canister was then interpolated to a 15-min interval flow rate using linear interpolation function (Igor 6.3.7.2). The total N2O flow rate of the 16 canisters (QN2O) was used for the CH4 emission rate calculation. 3) Following the procedure described in Bai (2010), Griffith et al. (2008), and Jones et al. (2011), the herd emission rate of CH4 was calculated (Eq.3): QCH4 =  $QN2O^*(\Delta CH4/\Delta N2O)$ \*(MCH4/MN2O)/Nanimal (3) Where QCH4 is the CH4 emission rate, g head-1 h-1, QN2O is the integrated N2O flow rate of total canisters in the animal backpacks, determined by mass loss of N2O at canister temperature T and release time t, g h-1, is multiplied by 24 to calculated g head-1 d-1. The  $\triangle$ CH4 and  $\triangle$ N2O parameters are the CH4 and N2O concentration enhancements (above the local background level) measured downwind of the animal pen using the OP-FTIR spectrometers, MCH4 is the molecular mass of CH4, 16 g mol-1, MN2O is the molecular mass of N2O, 44 g mol-1, Nanimal is animal number, 16. I 78:define exactly what Qch4 is, with units. QCH4 refers CH4 emission rate, g head-1 h-1.

1110: could the data collected while the animals were absent be shown, to demonstrate the noise/sensitivity? This provides a neat control period with zero emission. We used this period to calibrate OP-FTIR and OP-laser. 1) During the study we collected a number of air samples using volumetric flasks (600 mL). Samples were spaced along each measurement path and taken when animals were absent from the pen. These samples were later analysed in the laboratory using a closed-path FTIR spectrometer (Griffith, 1996) and the CH4 and N2O values were used to cross-calibrate the two OP-FTIR sensors. 2) Samples were also collected (20 mL) into evacuated 12mL vials (Exetainer®, Labco Ltd., Ceredigion, UK) at 2 min intervals to get 15-min average concentrations for the period from 9:15 - 9:30 am. This was a period when the cattle were not in the paddock. The samples were analyzed by the gas chromatography (GC)(Agilent 7890A, Wilmington, USA) at the University of Melbourne laboratory. Three positions were sampled: 1) directly west of the paddock along the laser/FTIR line, 2) near the laser, southwest of the paddock, and 3) far south of the paddock along the southerly laser line. Winds were from light and from the east. We hoped the CH4 and N2O concentrations at these positions would be similar (as cattle were absent) and would provide the basis for calibration of the lasers and FTIRs. The sample analysis showed that CH4 was elevated at the location immediately downwind of the paddock (CH4 = 1.88 ppm). The other two locations had average values of 1.75 and 1.80 ppm - reasonable background levels. Therefore these two concentration measurements can be averaged (C = 1.77 ppm) and used to calibrate the lasers and FTIR on the paths not downwind of the paddock: the 1.77 ppm value was used to calibrate the scanner laser on the south-pointed path, and to calibrate the SW FTIR on the east-pointed path, and the NE FTIR on the south-pointed path. This information are described in the revised manuscript. I 131: how big is the sensor drift? Is this a large uncertainty? The drift was between 5% to 15% with larger drift of OP-Laser.

I 144: touchdowns - the whole Lagrangian particle idea needs to be explained. We added the information of IDM technique in the revised manuscript. Pages 6-7, lines

203-210. "Herd CH4 emissions were calculated using the IDM technique (Flesch, Wilson, Harper, Crenna and Sharpe, 2004). This micrometeorological technique estimates emissions based on the enhancement of CH4 measured downwind of the animal pen. The link between the concentration enhancement and the pen emission rate is calculated using an atmospheric dispersion model. The freely available software Wind-Trax (www.thunderbeachscientific.com) is used for that calculation. WindTrax combines a backward Lagrangian stochastic dispersion model with mapping software and takes as input: the upwind and downwind CH4 concentration measurements, wind information from a sonic anemometer, and a map of the pen and gas sensor locations. General information on WindTrax applications is given in Flesch and Wilson (2005)." I 146: spec.max? Spec. max refers spectral signal intensity.

I 149: why was the diel cycle used? What is driving the diel cycle in methane emission? Production rate should be constant, but emission will be affected by feeding behaviour, Or is this cycle a measurement artefact? This needs explaining. Other gap-filling methods might be better e.g. smoothers such as GAMs. We have several literature examples where emission rate observations have been grouped by time-of-day to come up with an ensemble 24-hour emission curve. For example, Bai et al., (2015), Loh et al., (2008) and Laubach et al. (2013). We used Generalized Additive Models (GAM) fitted to the time series of gas emission to impute missing measurements (see Fig. A1 below) (Bai et al., 2020). Note that longer periods of missing data combined with a higher residual error component and a higher wiggliness in the GAM smoother inflates estimate uncertainty for the IDM-Laser method. Insert Figure A1.

I 171: Yield needs to be explained. Methane yield refers g CH4 kg-1 dry matter intake (DMI).

I 216: This question is not answered very clearly. The authors seem to find no reason, except the IPCC value should perhaps have wider uncertainty bounds on it. Agree. We changed the sentence to be "This suggests that IPCC estimates may have larger uncertainties." Page 11, line 332.

References Bai, M., 2010. Methane emissions from livestock measured by novel spectroscopic techniques. PhD Thesis Thesis, University of Wollongong, 303 pp. Bai, M., Flesch, K.T., Trouvé, R., Coates, T.W., Butterly, C., Bhatta, B., Hill, J. and Chen, D., 2020. Gas Emissions during Cattle Manure Composting and Stockpiling. J. Environ. Qual., 49: 228-235. Bai, M., Flesch, T., McGinn, S. and Chen, D., 2015. A snapshot of greenhouse gas emissions from a cattle feedlot. J. Environ. Qual., 44(6): 1974-1978. Flesch, K.T., Baron, V., Wilson, J., Griffith, D.W.T., Basarab, J. and Carlson, P., 2016. Agricultural gas emissions during the spring thaw: Applying a new measuremnt technique. Agric. Forest Meteorol., 221: 111-121. Flesch, T.K. and Wilson, J.D., 2005. Estimating tracer emissions with a backward Lagrangian stochastic technique. In: J.L. Hatfield, J.M. Baker and M.K. Viney (Editors), Micrometeorology in agricultural systems. American Society of Agronomy, Inc. Crop Science Society of America, Inc. Soil Science Society of America, Inc., U.S.A., pp. 513-531. Flesch, T.K., Wilson, J.D., Harper, L.A., Crenna, B.P. and Sharpe, R.R., 2004. Deducing ground-to-air emissions from observed trace gas concentrations: A field trial. J. Appl. Meteorol., 43: 487-502. Griffith, D.W.T., 1996. Synthetic calibration and quantitative analysis of gas-phase FT-IR spectra. Appl. Spectrosc., 50: 59-70. Griffith, D.W.T., Bryant, G.R., Hsu, D. and Reisinger, A.R., 2008. Methane emissions from free-ranging cattle: comparison of tracer and integrated horizontal flux techniques. J. Environ. Qual., 37: 582-591. Jones, F.M., Phillips, F.A., Naylor, T. and Mercer, N.B., 2011. Methane emissions from grazing Angus beef cows selected for divergent residual feed intake. Anim. Feed Sci. Tech., 166: 302-307. Laubach, J., Bai, M., Pinares-Patiño, C.S., Phillips, F.A., Naylor, T.A., Molano, G., Cárdenas Rocha, E.A. and Griffith, D.W.T., 2013. Accuracy of micrometeorological techniques for detecting a change in methane emissions from a herd of cattle. Agr. Forest Meteorol., 176(0): 50-63. Loh, Z., Chen, D., Bai, M., Naylor, T., Griffith, D., Hill, J., Denmead, T., McGinn, S. and Edis, R., 2008. Measurement of greenhouse gas emissions from Australian feedlot beef production using open-path spectroscopy and atmospheric dispersion modelling. Aust. J. Exp. Agr., 48: 244-247. McGinn, S.M., Flesch, T.K., Coates, T.W., Charmley, E., Chen, D., Bai, M. and Bishop-Hurley,

G., 2015. Evaluating dispersion modelling options to estimate methane emissions from grazing beef cattle. J. Environ. Qual., 44: 97-102. McGinn, S.M., Turner, D., Tomkins, N., Charmley, E., Bishop-Hurley, G. and Chen, D., 2011. Methane Emissions from Grazing Cattle Using Point-Source Dispersion. J. Environ. Qual., 40(1): 22-27.

Fig. 1. Figure A1: Time series of CH4 emissions measured using the Tracer-Ratio, IDM-FTIR, and IDM-Laser methods. Black dots show the 15 minutes measurements. The solid black line shows the mean value of gas

---

## Author Response (AR1)

Dear reviewer (Anonymous Referee #1),

Re: Revision of manuscript Number: AMT-2020-445, Title: **Beef cattle methane emissions measured with tracer-ratio and inverse dispersion modelling techniques**

We thank your positive feedback on the manuscript. We have addressed the comments thoroughly, our response to every issue raised is given point by point in **blue text** below.

** Abstract **
1. Should be reconsidered based on the remark of the other sections. I would suggest to add the details of the experiment's management, e.g. duration of the trial, pointing that was an experimental pen. Adding the type of the FTIR and the Laser used can be useful to understand the method used. I would suggest also to add the uncertainty of the cumulative flux.

The information for the trial period and experimental pen as suggested has been added in the revised manuscript. The details of the type of instrument (OP-FTIR and OP-Laser), tracer-ratio and inverse dispersion modelling technique have been described in the Materials and Methods section, therefore we did not add this information in Abstract.
Since we already documented the uncertainty in the average daily emission rate, no additional information in the cumulative uncertainty is needed.

**Introduction**
2. The introduction is somewhat lacking and should be extended. Moreover, some elements missing to understand the novelty of this work compared to literature. First, I suggest adding some data from the inventories related to the contribution of the livestock to GHG emissions in Australia and the World. This Reviewer suggest a series of questions to be answered with the purpose to improve the introduction section: What is CH4 and how it affects the climate? Where does CH4 emissions come from? How agriculture (and livestock) contributes to CH4? Which are the most prominent mitigation options?

We have added the information in the revised manuscript.
"Agriculture is the main source of anthropogenic methane ($CH_4$) emitted to the atmosphere, which includes emissions from ruminants, rice agriculture, waste treatment, and biomass burning (Solomon et al., 2007). Methane is an important greenhouse gas (GHG) with a global warming potential that is 28 times that of carbon dioxide ($CO_2$) in a 100 year time (Myhre et al., 2013). Enteric $CH_4$ from livestock is a major source of GHG emissions. A significant effort is being made to mitigate these emissions through diet modification feed supplements, farm management, grazing strategies, and animal breeding (Min et al., 2020; Vyas et al., 2018); with ruminant nutritional management strategies seen as the most direct impact mitigation option (Cottle et al., 2011)." Page 1, lines 28-35.

3. Secondly, the section describing the different available methods and techniques to measure enteric CH4 is, in my opinion, poor and should be improved. I suggest to add examples from literature, which can be used also in the discussion section (this Reviewer add here a non-exhaustive list: Felber et al., 2015; Dengel et al., 2011; Lockyer and Jarvis, 1995; Grainger et al., 2007; Laubach et al., 2008; Todd et al., 2014). The strengths and weaknesses of each measurement method or technique should be stressed in order to defend the type of methodologies used by this study. I suggest also adding some experience about the tracer-ratio technique, since it is

defined as "true" in this paper (L 69).

It is beneficial to give some broad context to different methods, mainly to support our (critical) assertion that the tracer-ratio technique is the most accurate method in principle. The information has been added in the revised manuscript. Page 2, lines 38-51.

"On-farm enteric emissions have been measured using three main techniques. 1) Portable respiration hoods for tethered and non-tethered animals (Garnsworthy et al., 2012; Zimmerman and Zimmerman, 2012) directly measure the gas concentration of incoming and exhaust air from individual animals. However, this technique limits the animal's movements, requires intensive training for animals and labor, and it does not account for emissions from the animal rectum. 2) Tracer-ratio gas releases from the animal (Johnson et al., 1994), such as $SF_6$ (Grainger et al., 2007), assumes the tracer gas and the emitted $CH_4$ have similar transport paths, so that a tracer measurement can establish the $CH_4$ emission rate. This is a simple technique, but there are challenges with logistics and handling animals similar to the respiration hood technique. 3) Micrometeorological techniques are typically considered a herd-scale measurement, where the emission rate is calculated from the measurement of enhanced gas concentrations downwind of an animal herd (Harper et al., 2011), and these include the mass balance technique (Laubach et al., 2008; Lockyer and Jarvis, 1995), eddy covariance (Dengel et al., 2011; Felber et al., 2015), and inverse dispersion techniques (Flesch et al., 2005; Todd et al., 2014). The main advantage of micrometeorological techniques is that they do not interfere with the animals or the environment."

Following the suggestions of adding some experience about the tracer-ratio technique, the information can be found in the Materials and Methods section. Page 5, lines 169-192.

4. Third, the novelty of this study. If the IDM technique has been already applied to perform the quantification of CH4 from grazing animals, I would encourage adding these information and stressing how the work you are presenting has some novelty (technical, methodological, or environmental conditions) compared to the literature and previous studies (i.e. Bai 2010). Finally, why do you compared two different concentration measurement tools?

The reviewer asks a good question -- why include the laser system in the method comparison? It is true that the laser results are not needed to compare the IDM and Tracer Ratio techniques. But one reason for including the laser results is to justify the relatively high emission rates found with the FTIR measurements (in relation to the IPCC estimates and expectations). The fact that an independent system (with emissions calculated independently from the FTIR system) gave similar results supports the unexpectedly high emissions rates found by the FTIR system.

**Materials and Methods**
5. This section should be reorganised. I suggest providing a detailed section of the laser and FTIR, along with their working principle and field setup, just below the Experimental Design section, then describe the two methods (tracer-ratio and IDM). The information about the FTIR and Laser is scattered and not well organised, disadvantaging readability and understanding. See also the comments #10 to 12; and #22. I suggest including here the details of the calculation of the losses made with IPPC's guidelines (Table 1); comment #21
Agree. As suggested, we reorganized the Materials and Methods section in the revised manuscript. page 8, lines 249-251.We also added the information of methane emission calculated using IPCC recommendation based on the dry matter intake (DMI).

"Following IPCC (2006) recommendation, $CH_4$ emission using were also calculated based on DMI (Eq. 10.21). This assumes $CH_4$ energy content = 55.65 MJ (kg $CH_4$)$^{-1}$, DMI energy content = 18.45 MJ (kg DMI)$^{-1}$, and $CH_4$ conversion factor $Y_m$ = 6.5%."

**Experimental design**

6. I suggest adding (here or in the Results section, see comment #19) more details about the experimental site, as the meteorological variables measurements from the weather station (e.g. rain, temperatures, wind direction and speed). This will help the reader to understand the validity of the measurements (e.g. wind direction), this particular environment and, of course, the results.

Agree with the reviewer. Figure 2 with the ambient temperature, wind speed, wind direction, z/L and $u_*$ is added to the revised manuscript.

[Figure]

**Figure 2 Ambient temperature (Airtemp), wind speed, wind direction was measured during the study. Atmospheric stability parameter (z/L) and wind friction velocity ($u_*$) are also plotted.**

7. Can the Authors detail here more about the dejections management during the experiment in order to better understand the field set-up and neglecting further sources of methane?

We have added a sentence in the revised manuscript "There were no other cattle or animal manure storages nearby during the study …" page 2, lines 61-62.

8. L 54-55 I would suggest to add a reference to the part of the tracer description (2.2.1)

Agree with reviewer's suggestion. A reference has been added to the revised manuscript.

9. Figure 1. I warmly suggest re-making the picture with a proper scale. This will help better understating the field setting and the distances of the probes (and the weather station) from the fences.

Agree. we have modified the figure (Fig. 1).

[Figure]

**Figure 1: Schematic layout of the experimental site, showing an animal pen in the center, two OP-FTIR systems (blue dashed lines) and the OP-Laser system (red dashed lines). Two feeding troughs (brown squares) were on both sides of the pen, and one water trough (brown circle) was on the north of the pen. A weather station (green triangle) was 50 m away from the SW corner of the animal pen.**

\* Methodologies \* \* Tracer-ratio technique (N2O Tracer) \*
10. I would suggest to explain with more detail and in a few line how is the principle of this method. L75, please explain what is QCH4 in the text.
The information has been added in the Introduction, as well as the methodologies section. Pages 5-6, line 169-192.

The calculation for each pressurized canister $N_2O$ flow rate follows three steps:
1) The $N_2O$ flow rate of each canister was calculated following Bai (2010) (Eq.1):
$$Q_{N2O}(t) = Q_0 + \alpha\, T(t) \qquad\qquad (1)$$
Where $Q_{N2O}(t)$ is the individual canister flow rate (g h$^{-1}$) at temperature T (°C), t = time, T = temperature °C at time (t), $Q_0$ is a constant canister flow rate at temperature 0°C, g h$^{-1}$, $\alpha$ is the $N_2O$ flow rate temperature dependent factor, g h$^{-1}$°C$^{-1}$. The temperature was measured at 5-min intervals.

2) The integrated $N_2O$ flow rate over the total release time (RT, ~24h) equals the mass loss of $N_2O$ gas ($\Delta m_{N2O}$, g) (Eq.2):

$$Q_0 = (\Delta m_{N2O} / RT) - (\Sigma (\alpha \, T \, (t)))/RT \qquad (2)$$

Where $\Delta m_{N2O} = WN_2O_{start} - WN_2O_{end}$

The mass loss of $N_2O$ was determined by the initial and the end weight of the canister (g), $WN_2O_{start}$, $WN_2O_{end}$, respectively. The integrated $N_2O$ flow rate of each canister was then interpolated to a 15-min interval flow rate using linear interpolation function (Igor 6.3.7.2). The total $N_2O$ flow rate of the 16 canisters ($Q_{N2O}$) was used for the $CH_4$ emission rate calculation.

3) Following the procedure described in Bai (2010), Griffith et al. (2008), and Jones et al. (2011), the herd emission rate of $CH_4$ was calculated (Eq.3):

$$Q_{CH4} = Q_{N2O} * (\Delta CH_4 / \Delta N_2O) * (M_{CH4}/M_{N2O})/N_{animal} \qquad (3)$$

Where $Q_{CH4}$ is the $CH_4$ emission rate, g head$^{-1}$ h$^{-1}$, $Q_{N2O}$ is the integrated $N_2O$ flow rate of total canisters in the animal backpacks, determined by mass loss of $N_2O$ at canister temperature T and release time t, g h$^{-1}$, is multiplied by 24 to calculated g head$^{-1}$ d$^{-1}$. The $\Delta CH_4$ and $\Delta N_2O$ parameters are the $CH_4$ and $N_2O$ concentration enhancements (above the local background level) measured downwind of the animal pen using the OP-FTIR spectrometers, $M_{CH4}$ is the molecular mass of $CH_4$, 16 g mol$^{-1}$, $M_{N2O}$ is the molecular mass of $N_2O$, 44 g mol$^{-1}$, $N_{animal}$ is animal number, 16.

11. Can the Authors add the details of the producer of the FTDIR, the measurement range (to justify also lines L112-113 and L142-145), the uncertainty and sensitivity (to justify lines L162, L178), and all the technical parameters that can help to characterise this measurement. Could the Authors detail where the measurements were recorded ?

Yes. The details of the information have been added in the revised manuscript. Pages 3-4, lines 91-128.

**2.2 Concentration sensors**
**2.2.1. OP-FTIR**
Atmospheric concentrations of $CH_4$ and $N_2O$ were measured upwind and downwind of the cattle pen using two open-path Fourier transform infrared (OP-FTIR) spectrometers. OP-FTIR can quantify a wide range of real-time gas concentrations simultaneously with high resolution (Smith et al., 2011). The details of the OP-FTIR system used in this study can be found in Bai (2010) and Paton-Walsh et al. (2014). Briefly, the modulated infrared (IR) beam from the Bruker IRcube spectrometer (Matrix-M IRcube, Bruker Optics, Ettlingen, Germany) is transferred through the optics to a modified Meade Schmidt-Cassegrain telescope (25.4 cm diameter, Model LX200R, Meade Instrument Corp., Irvine, California, USA) and a secondary mirror, and diverged to 250 mm parallel beam and extended to a distant retro reflector (up to 500 m from the spectrometer) (PLX Industries, Deer Park, New York, USA). The parallel beam is then reflected by the retro reflector and returned to a Mercury Cadmium Telluride (MCT) detector (Infrared Associates Inc., Stuart, Florida, USA) where temperature is controlled by a Stirling cycle mechanical refrigerator cooling system (-196 °C) (Ricor K508, Salem, New Hampshire, USA), as described further in (Bai, 2010). A Zener-diode thermometer (type LM335) and a barometer (PTB110, Vaisala, Helsinki, Finland) provide real-time ambient temperature and pressure data (at the same height of the measurement path) for the analysis of the measured spectra. The spectrometer is operated at 1 cm$^{-1}$ resolution, and one spectrometer scan takes approximately 4 secs (13 scans min$^{-1}$). For acceptable signal to noise ratios, a minimum measurement period of 1 min is required. The measured spectra are quantitatively analyzed using the MALT analysis program and a nonlinear least squares fitting procedure described in Griffith (1996), based on the reference spectra from the molecule absorption databases (HITRAN) (Rothman et al., 2009). The best fitted spectrum is used to retrieve the line-average gas concentrations of $CH_4$ and $N_2O$ over the measurement path. The sensitivity of the OP-FTIR units for $CH_4$ and $N_2O$ is 1 part

per billion (ppb), corresponding to 2 and 0.4 ppb for a 100 m path, respectively. To achieve good spectra, parameters including instrument field-of-view (FOV), spectral signal intensity (spec. max), and the residual spectrum between the measured and modelled spectra (RMSresid)are examined. A software "Spectronous" (Ecotech, Knoxfield, Victoria Australia) automatically controls spectrometer, sample collecting, spectrum analysis, data logging and display of the calculated concentrations in real time, together with ambient pressure and temperature."

12. Can the Authors add some information about the close-path FTIR used in laboratory in this section, or in the section of the concentration measurements (see comment #5)?
The close-path FTIR analysis was operated by a laboratory staff, the information required by the reviewer is out of this study.

13. This section should represents one of the main methodological part of the paper and, I retain, it can be improved. I suggest adding the principle behind the backward application of the short-range Lagrangian dispersion model used in the study (equation, number of trajectories used and principles of the MOST). This will improve the understanding of the scientific ground, the application of this technique in the case study presented here and better understand the Equation 1. Any reference to other study using IDM in the short range is recommended.
Agree. we have added the information in the revised manuscript. Pages 6-7, lines 203-210.

"Herd $CH_4$ emissions were calculated using the IDM technique (Flesch et al., 2004). This micrometeorological technique estimates emissions based on the enhancement of $CH_4$ measured downwind of the animal pen. The link between the concentration enhancement and the pen emission rate is calculated using an atmospheric dispersion model. The freely available software WindTrax (www.thunderbeachscientific.com) is used for that calculation. WindTrax combines a backward Lagrangian stochastic dispersion model with mapping software and takes as input: the upwind and downwind $CH_4$ concentration measurements, wind information from a sonic anemometer, and a map of the pen and gas sensor locations. General information on WindTrax applications is given in Flesch and Wilson (2005)."

14. Please, detail how the roughness length was calculated (reference or equation, and the results). Furthermore, can the Authors detail if they're using a constant or a variable z0.
A reference (Garratt, 1992) is added. For z0, state that it is a variable inferred from the sonic anemometer measurements, as described in Flesch et al. (2004).

15. L139-142. Can you address why these thresholds were imposed for this case study and why these are different from Flesch et al 2005? Please, refer here to the methodological part requested in the comment #13. Furthermore, how many "15-mins" data were excluded from the dataset with these thresholds and in which part of the day?
The meteorological (bLS) thresholds used in Flesch et al. (2005) were not presented as universal. Other studies have used different values.

"Over the seven-study days, emissions were measured during 90% of the ensemble 24 h day (i.e., 86 of the 96 possible 15-min periods)." The data were discarded were due the filtering criteria "In the IDM analysis we followed the procedure of Flesch et al. (2005) to remove error-prone intervals when either $u_* < 0.15$ m s$^{-1}$, $|L| < 5$ m, $z0 < 0.9$ m, or when the fraction of WindTrax trajectory touchdowns inside the pen source covered < 10% of the pen area. Intervals were also removed when the concentrations measured by the OP-FTIR or the laser corresponded to low signal levels: i.e., FOV < 35, RMSresid < 0.2%, spec.max was < 0.25, in the spectral region of 2200 cm$^{-1}$ for the OP-FTIR, or the

light level reported by the laser fell outside the 2000–13000 range, or the laser quality parameter $R^2$ < 0.97. "

16. L 143. Please, explain what "spec.max" stands for.
Spec.max stands for spectral signal intensity, which has been described in the revised manuscript. Page 4, line 114.

17. I would suggest to rewrite this part more clearly, giving some reference to other studies which use the same calculation. This will greatly help the reader. A gap-filling procedure has been used? Pleas add these details.
We have several literature examples where emission rate observations have been grouped by time-of-day to come up with an ensemble 24-hour emission curve. For example, Bai et al., (2015), Loh et al., (2008) and Laubach et al. (2013).

We also added the information for gap-filling procedure on page 7, lines 243-246.
"We used Generalized Additive Models (GAM) fitted to the time series of gas emission to impute missing measurements (Bai et al., 2020). The time series of gas emission and associated GAM fit for each measurement method are shown in Appendices (Fig. A1)."

18. Were the periods when the animals were not in the pen excluded from the measurement dataset? This point should be better described.
We calibrated the measurement with air sample measurement during this period when animals were absent. See page 4, lines 194-197.

"Samples were spaced along each measurement path and taken when animals were absent from the pen. These samples were later analyzed in the laboratory using a closed-path FTIR spectrometer (Griffith, 1996) and the $CH_4$ and $N_2O$ values were used to cross-calibrate the two OP-FTIR sensors."

** Results **
*Climate condition*

19. This part should be improved and extended. I would suggest adding a figure with the dynamics of air temperature, wind speed and rain, at least. Moreover, I warmly recommend to add a figure with the trends of u* and the turbulence parameter z/L.
see the response to # 6.

20. To better understand the measurement performed, given that two different methods are compared in this study (Laser and FTIR), it might be interesting to evaluate the concentrations observed over time by the two systems and by the tracer, before evaluating the final daily cumulative emissions. I suggest to provide these results.
Agree. As suggested by the reviewer, we have plotted the enhanced concentrations of $CH_4$ and $N_2O$ from OP-FTIR and the enhanced concentration of $CH_4$ from OP-Laser system.

[Figure]

**Figure 3. The concentration enhancement of N₂O and CH₄ from OP-FTIR and CH₄ from OP-Laser over the measurement period of 14−21 February 2013.**

21. Table 1. I suggest putting the measurement uncertainty for each of the measurements. I would also suggest removing the reference (Charmley) from the table and keep it exclusively in discussions section along with the other sources cited to defend your findings. Furthermore, I would better explain the calculation with the IPCC's guidelines in materials and methods (see comment #5).
Yes, agree. We removed the reference from the table and added the calculation information in the Materials and Methods section.

22. L161-163. What about the sensitivity of the laser source?
We added the information in the revised manuscript. Page 5, lines 144-145.

"The sensitivity of the laser units is 1 part per million-metre (ppm-m), corresponding to 10 ppb for a 100-m path."

* The inverse-dispersion modelling (IDM) emissions *
23. L175. I would suggest to detail better what "low wind speed" means for the Authors. Or, if these percentages are comprehensive of the periods not considered because of the MOST conditions failure (L139-142)?
Low wind speed refers to the wind velocity ($u_*$) is less than 0.15 m s$^{-1}$ when the MOST conditions failed. This has been described in the text "In the IDM analysis we followed the procedure of Flesch et al. (2005) to remove error-prone intervals when either $u_* < 0.15$ m s$^{-1}$, $|L| < 5$ m, $z_0 < 0.9$ m, or when the fraction of WindTrax trajectory touchdowns inside the pen source covered < 10% of the pen area."

24. I cannot see any comparison about the "sensitivity" of the two sensors. I suggest to address this part on the Materials and Methods section (see comment #5) and in the results (comment #19).
See the response to #22

25. The lowest emission value is at 9 am, the time when the animals left the pen. How did this event affect the dataset? Are these gaps filled and how ?

Number of 15-min observations is used to create the average (i.e. 4 obs per hour X 7 days= ~28 is the maximum). The cattle were scheduled to be out at around 9 am, and it took about 15 to 30 min to change the canisters. There was maximum 4 observation an hour and minimum 1 observation.

26. I would warmly suggest to insert a further figure about the trend of 15-mins emissions over the 7 days of measurement. This will give the real picture of the dataset, without the period of failures (technical), filtered because of the MOST failure. We used Generalized Additive Models (GAM) fitted to the time series of gas emission to impute missing measurements (Bai et al. 2020).The time series of gas emission and associated GAM fit for each measurement method are shown in Appendices Appendix A (Fig. A1).

[Figure]

**Figure A1: Time series of CH₄ emissions measured using the Tracer-Ratio, IDM-FTIR, and IDM-Laser methods. Black dots show the 15 minutes measurements. The solid black line shows the mean value of gas emission estimated from a GAM fit to the measurement data. The shaded area represents the 95% credible intervals of the mean gas emission from the GAM fit (i.e., it contains 95% of the potential mean values of gas emission at a given time).**

Furthermore, we have assumed that the animals were well adjusted to the site and feeding regime (ad libitum), and we expect that relatively consistent emission rates. Given the discontinuous nature of the emission measurements, it is better to look at averaged emission rates.

27. Figure 2. IDM-FTIR does not have the measurement at 11pm
We thank the reviewer for pointing out the mistake. The figure has been modified (Figure 4).

[Figure]

**Figure 4: Ensemble 24-h diel CH₄ emission pattern measured by IDM-Laser, IDM-FTIR, and Tracer-Ratio method (hourly values based on 7-d of measurements). Error bars denote the standard error of mean.**

28. The discussions should be better set up and expanded with other literature studies to defend the validity of the measurement, i.e. defending that the conditions of the experiment were always suitable for the application of the IDM. It seems that the reliability of the IDM method is related only to the final cumulative emissions (Table 1). In order to define that the source was homogeneous, and therefore the monitoring of the animals is not needed, as stated, further results from this study - or results from other studies -should be provided.
In this study the evaluation standard for IDM is agreement with the Tracer-Ratio. We assume, with good reason, that the Tracer-Ratio approach is the most accurate means of measuring emissions in ambient conditions. In this case, agreement with other studies (using other methods, in other animal situations) is not useful.

29. Referring only to the method of the IPCC guidelines is, in my opinion, limited. I would suggest broadening the discussions with other case studies, reporting their characteristics and results to make the measurement more robust (e.g. references cited online 226).
The introduction of an IPCC emission value is a recognition that our measured emission rates are high. This gives some interesting context to our measurements (and I think it is something the audience would question), but it is not crucial to our primary objective of comparing IDM and Tracer-Ratio.

30. The conclusions, with respect to the use of IDMs, should be much more cautious given that this is an experiment of only 7 days, performed in micrometeorological conditions not detailed in the paper, without a real defence of the validity of the application of the method itself (homogeneity of the source).
While seven days may not be sufficient to document long-term cattle emissions, seven days of near-continuous measurements is not an insignificant period when comparing micrometeorological

techniques. One should not discount the very close agreement over a range of meteorological and animal position conditions.

results that implicitly include the impact of animal positions. This indicates that IDM studies like ours can use the much simpler approach where the whole paddock is treated as a gas source, and animal positions need not be monitored. This seems to confirm a similar finding from McGinn et al. (2015). The effect of this simplification on measurement accuracy is likely to depend on animal density and the size of the paddock. For example, the measurement of a small number of animals in a large paddock is likely to be very sensitive to the exact animal positions. But in the modest sized paddock studied here (and in McGinn et al., (2015)) this is not the case."

Given our experiment configuration, constant easterly winds, absence of light winds, and good quality data that 80-90% of measurement data is useful and sufficient to calculate the fluxes, we are confident with our results based on 7-day measurement.

Specific points:

l 72: the tracer-ratio technique is indeed simple, but cannot be considered "true". The uncertainty associated with its estimates needs to be quantified. The point is presumably that these uncertainties are smaller than the IDM method, but this needs to be demonstrated. E.g. how predictable is the N2O release rate? The authors say this has to be corrected for temperature dependency, but presumably this is established in lab tests?

The canister pressure is correlated with the ambient temperature. Prior to the field study, temperature dependent factor associated with canister release rate was tested in the laboratory, and $CO_2$ was used as the tracer gas due to its low cost. $CO_2$ flow rate was tested at a range of controlled temperature to determine the $CO_2$ release rate temperature dependent factor. Because $N_2O$ and $CO_2$ have the same molecular mass, we assumed that $N_2O$ and $CO_2$ disperse in a similar path in the atmosphere and have the same characteristic temperature dependence.  The $N_2O$ flow rate calculation consists of three steps:

1) The $N_2O$ flow rate of each canister was calculated following Bai (2010) (Eq.1):

$$Q_{N2O}(t) = Q_0 + \alpha\, T(t) \qquad\qquad (1)$$

Where $Q_{N2O}(t)$ is the individual canister flow rate (g h$^{-1}$) at temperature T (°C), t = time, T = temperature °C at time (t), $Q_0$ is a constant canister flow rate at temperature 0°C, g h$^{-1}$, $\alpha$ is the $N_2O$ flow rate temperature dependent factor, g h$^{-1}$ °C$^{-1}$. The temperature was measured at 5-min intervals.

2) The integrated $N_2O$ flow rate over the total release time (RT, ~24h) equals the mass loss of $N_2O$ gas ($\Delta m_{N2O}$, g) (Eq.2):

$$Q_0 = (\Delta m_{N2O}/RT) - (\Sigma\,(\alpha\, T(t)))/RT \qquad\qquad (2)$$

Where $\Delta m_{N2O} = WN_2O_{start} - WN_2O_{end}$

The mass loss of $N_2O$ was determined by the initial and the end weight of the canister (g), $WN_2O_{start}$, $WN_2O_{end}$, respectively. The integrated $N_2O$ flow rate of each canister was then interpolated to a 15-min interval flow rate using linear interpolation function (Igor 6.3.7.2). The total $N_2O$ flow rate of the 16 canisters ($Q_{N2O}$) was used for the $CH_4$ emission rate calculation.

3) Following the procedure described in Bai (2010), Griffith et al. (2008), and Jones et al. (2011), the herd emission rate of $CH_4$ was calculated (Eq.3):

$$Q_{CH4} = Q_{N2O} * (\Delta CH_4/\Delta N_2O) * (M_{CH4}/M_{N2O})/N_{animal} \qquad\qquad (3)$$

Where $Q_{CH4}$ is the $CH_4$ emission rate, g head$^{-1}$ h$^{-1}$, $Q_{N2O}$ is the integrated $N_2O$ flow rate of total canisters in the animal backpacks, determined by mass loss of $N_2O$ at canister temperature T and release time t, g h$^{-1}$, is multiplied by 24 to calculated g head$^{-1}$ d$^{-1}$. The $\Delta CH_4$ and $\Delta N_2O$ parameters are the $CH_4$ and $N_2O$ concentration enhancements (above the local background level) measured downwind of the animal pen using the OP-FTIR spectrometers, $M_{CH4}$ is the molecular mass of $CH_4$, 16 g mol$^{-1}$, $M_{N2O}$ is the molecular mass of $N_2O$, 44 g mol$^{-1}$, $N_{animal}$ is animal number, 16.

l 78:define exactly what $Q_{ch4}$ is, with units.
$Q_{CH4}$ refers $CH_4$ emission rate, g head$^{-1}$ h$^{-1}$.

l 110: could the data collected while the animals were absent be shown, to demonstrate the noise/sensitivity? This provides a neat control period with zero emission.
We used this period to calibrate OP-FTIR and OP-laser.
1) During the study we collected a number of air samples using volumetric flasks (600 mL).  Samples were spaced along each measurement path and taken when animals were absent from the pen. These samples were later analysed in the laboratory using a closed-path FTIR spectrometer (Griffith, 1996) and the $CH_4$ and $N_2O$ values were used to cross-calibrate the two OP-FTIR sensors.
2) Samples were also collected (20 mL) into evacuated 12mL vials (Exetainer®, Labco Ltd., Ceredigion, UK) at 2 min intervals to get 15-min average concentrations for the period from 9:15 – 9:30 am.  This was a period when the cattle were not in the paddock.  The samples were analyzed by the gas chromatography (GC)(Agilent 7890A, Wilmington, USA) at the University of Melbourne laboratory. Three positions were sampled: 1) directly west of the paddock along the laser/FTIR line, 2) near the laser, southwest of the paddock, and 3) far south of the paddock along the southerly laser line.  Winds were from light and from the east.  We hoped the $CH_4$ and $N_2O$ concentrations at these positions would be similar (as cattle were absent) and would provide the basis for calibration of the lasers and FTIRs.

The sample analysis showed that $CH_4$ was elevated at the location immediately downwind of the paddock ($CH_4$ = 1.88 ppm).  The other two locations had average values of 1.75 and 1.80 ppm -- reasonable background levels.  Therefore these two concentration measurements can be averaged (C = 1.77 ppm) and used to calibrate the lasers and FTIR on the paths not downwind of the paddock: the 1.77 ppm value was used to calibrate the scanner laser on the south-pointed path, and to calibrate the SW FTIR on the east-pointed path, and the NE FTIR on the south-pointed path.

This information are described in the revised manuscript.

l 131: how big is the sensor drift? Is this a large uncertainty?
The drift was between 5% to 15% with larger drift of OP-Laser.

l 144: touchdowns - the whole Lagrangian particle idea needs to be explained.
We added the information of IDM technique in the revised manuscript. Pages 6-7, lines 203-210.
"Herd $CH_4$ emissions were calculated using the IDM technique (Flesch, Wilson, Harper, Crenna and Sharpe, 2004). This micrometeorological technique estimates emissions based on the enhancement of $CH_4$ measured downwind of the animal pen. The link between the concentration enhancement and the pen emission rate is calculated using an atmospheric dispersion model. The freely available software WindTrax (www.thunderbeachscientific.com) is used for that calculation. WindTrax combines a backward Lagrangian stochastic dispersion model with mapping software and takes as input: the upwind and downwind $CH_4$ concentration measurements, wind information from a sonic anemometer, and a map of the pen and gas sensor locations. General information on WindTrax applications is given in Flesch and Wilson (2005)."
l 146: spec.max?
Spec. max refers spectral signal intensity.

l 149: why was the diel cycle used? What is driving the diel cycle in methane emission?
Production rate should be constant, but emission will be affected by feeding behaviour,
Or is this cycle a measurement artefact? This needs explaining. Other gap-filling methods might be better e.g. smoothers such as GAMs.

We have several literature examples where emission rate observations have been grouped by time-of-day to come up with an ensemble 24-hour emission curve. For example, Bai et al., (2015), Loh et al., (2008) and Laubach et al. (2013).

We used Generalized Additive Models (GAM) fitted to the time series of gas emission to impute missing measurements (see Fig. A1 below) (Bai, Flesch, Trouvé, Coates, Butterly, Bhatta, Hill and Chen, 2020). Note that longer periods of missing data combined with a higher residual error component and a higher wiggliness in the GAM smoother inflates estimate uncertainty for the IDM-Laser method.

[Figure]

Figure A1: Time series of CH₄ emissions measured using the Tracer-Ratio, IDM-FTIR, and IDM-Laser methods. Black dots show the 15 minutes measurements. The solid black line shows the mean value of gas emission estimated from a GAM fit to the measurement data. The shaded area represents the 95% credible intervals of the mean gas emission from the GAM fit (i.e., it contains 95% of the potential mean values of gas emission at a given time).

l 171: Yield needs to be explained.
Methane yield refers g $CH_4$ $kg^{-1}$ dry matter intake (DMI).

l 216: This question is not answered very clearly. The authors seem to find no reason,

except the IPCC value should perhaps have wider uncertainty bounds on it.

Agree. We changed the sentence to be "This suggests that IPCC estimates may have larger uncertainties." Page 11, line 332.

**References**

[revised manuscript text omitted]

---

## Author Response (AR2)

Dear Associate Editor Famulari,

Thank you for the decision. We addressed your comments and made few changes throughout the correction version of manuscript.

Regards,

Mei